# LongPO: Long Context Self-Evolution of Large Language Models through Short-to-Long Preference Optimization

**Guanzheng Chen**[1,2,3,*] **Xin Li**[2,3,†] **Michael Qizhe Shieh**[1] **Lidong Bing**[4]

[1]National University of Singapore [2]DAMO Academy, Alibaba Group
[3]Hupan Lab, 310023, Hangzhou, China
[4]Shanda AI Research Institute
`gc.chen@u.nus.edu, xinting.lx@alibaba-inc.com`
`michaelshieh@comp.nus.edu.sg, lidong.bing@shanda.com`

## Abstract

Large Language Models (LLMs) have demonstrated remarkable capabilities through pretraining and alignment. However, superior short-context LLMs may underperform in long-context scenarios due to insufficient long-context alignment. This alignment process remains challenging due to the impracticality of human annotation for extended contexts and the difficulty in balancing short- and long-context performance. To address these challenges, we introduce LongPO, that enables short-context LLMs to self-evolve to excel on long-context tasks by internally transferring short-context capabilities. LongPO harnesses LLMs to learn from self-generated short-to-long preference data, comprising paired responses generated for identical instructions with long-context inputs and their compressed short-context counterparts, respectively. This preference reveals capabilities and potentials of LLMs cultivated during short-context alignment that may be diminished in under-aligned long-context scenarios. Additionally, LongPO incorporates a short-to-long KL constraint to mitigate short-context performance decline during long-context alignment. When applied to Mistral-7B-Instruct-v0.2 from 128K to 512K context lengths, LongPO fully retains short-context performance and largely outperforms naive SFT and DPO in both long- and short-context tasks. Specifically, LongPO-trained models can achieve results on long-context benchmarks comparable to, or even surpassing, those of superior LLMs (e.g., GPT-4-128K) that involve extensive long-context annotation and larger parameter scales. Our code is available at `https://github.com/DAMO-NLP-SG/LongPO`.

## 1 Introduction

Recent advancements in Large Language Models (LLMs) have revealed remarkable capabilities through extensive pretraining and subsequent alignment with human intentions. The alignment process, including methods such as Supervised Fine-Tuning (SFT) (Wei et al., 2022), Direct Preference Optimization (DPO) (Rafailov et al., 2023), and Reinforcement Learning from Human Feedback (RLHF) (Christiano et al., 2017; Ouyang et al., 2022; Stiennon et al., 2020), has effectively unleashed the potential of LLMs acquired during pretraining to achieve desired behaviors.

Although off-the-shelf alignment methods have made significant strides in short-context settings, their application to long-context situations remains challenging (Bai et al., 2024). First, the scarcity of high-quality, long-context annotated data poses a significant hurdle. Human annotation becomes impractical and less-reliable as context length increases (Dubey et al., 2024), while synthetic data generation using advanced LLMs lacks scalability and remains resource-intensive. Moreover, simply concatenating existing short-context datasets has been shown to yield unsatisfactory long-context performance (Liu et al., 2024b). Second, long-context alignment methods grapple with the balance

---

*This work was done during the internship of Guanzheng Chen at Alibaba DAMO Academy.
†Corresponding Author.

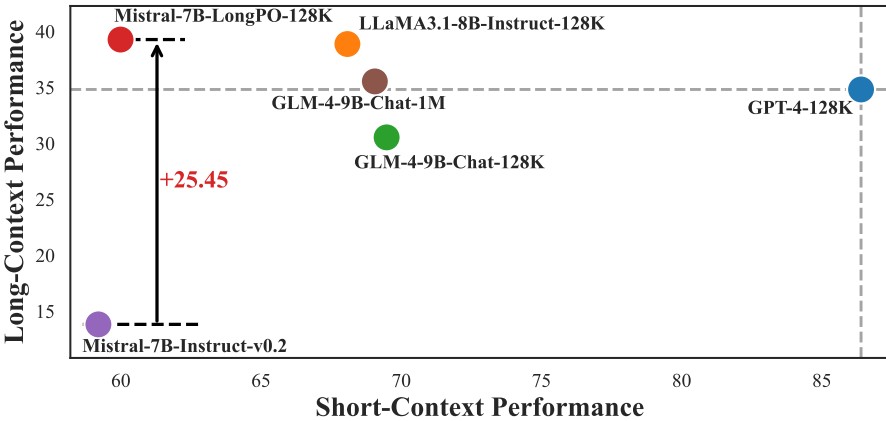

Figure 1: The comparison of long-context (∞Bench) and short-context (MMLU) performance among GPT-4-128K and smaller LLMs.

between preserving short-context proficiency and cultivating long-context capabilities (Liu et al., 2024b). For instance, the LLaMA-3.1 series incorporate merely 0.1% long-context data with over 99% short-context data during alignment to maintain the short-context performance (Liu et al., 2024b). This limited exposure to natural long-context data may result in insufficient alignment, potentially blocking the intrinsic long-context capabilities in LLMs.

The challenges of long-context alignment suggest that the full potential of LLMs may remain untapped for long-context tasks. As illustrated in Figure 1, even superior models such as GPT-4, which excel in short-context tasks, unexpectedly underperform in long-context scenarios. Interestingly, despite the much stronger short-context capabilities, GPT-4 is still inferior to LLaMA3.1-8B on long-context tasks. This disparity underscores the need for more effective long-context alignment methods to fully unleash the intrinsic power of LLMs across variable context lengths.

In this work, we posit that the capabilities, deeply ingrained during short-context pretraining and alignment, can be effectively transferred to longer contexts without external guidance. To this end, we introduce Short-to-**Long P**reference **O**ptimization (**LongPO**), to steer long-context alignment by injecting internal short-context preferences into long-context scenarios. Specifically, we propose to construct the preference data pairs by prompting the short-context LLM (e.g., Mistral-Instruct) with two inputs: (1) a long input comprising an instruction over a long document and, (2) a short input with the identical instruction over the relevant shortened chunk within the same document. We then designate the responses to short and long inputs as chosen and rejected responses, respectively. The short-to-long preference, i.e., the discrepancies between each paired response, reveal the capabilities and potentials cultivated during short-context alignment that may be diminished in under-aligned long-context scenarios. In order to bring forward the established capabilities, LongPO is utilized to optimize the model towards short-to-long preferences using DPO-style objectives upon long contexts. Furthermore, to maintain the short-context performance, we incorporate a *short-to-long constraint* in LongPO by applying Kullback-Leibler (KL) divergence between the response distributions to short and long inputs, respectively. This constraint, inspired by the KL constraint in RLHF (Ouyang et al., 2022; Stiennon et al., 2020), guides the policy model to minimize the deviation from its short-context output distribution when giving the long context during training. We found that this straightforward constraint largely enhances the retention of short-context performance after the long-context alignment.

We apply LongPO to Mistral-7B-Instruct-v0.2 (Jiang et al., 2023) and Qwen2.5-7B-Instruct, while iteratively extending their context lengths up to 512K, with the self-generated short-to-long preference data only. The experimental results demonstrate that LongPO, as a long-context alignment method, surpasses naive SFT and DPO by large margins (over 10 points) in both long- and short-context tasks. Notably, LongPO fully retains the performance of short-context LLMs after long-context alignment, whereas SFT and DPO yield substantial performance degradation (10~20 points on most tasks). In terms of long-context performance, LongPO largely improves the Mistral-7B-Instruct-v0.2 by 25.45 points on ∞Bench. Specifically, as depicted in Figure 1, the resulting model

is comparable with superior long-context LLMs at various scales (e.g., Mistral-7B-LongPO-128K of 39.27 vs. GPT-4-128K of 34.81 on ∞Bench), despite the latter often involving extensive continual training on hundreds of billions of tokens (Dubey et al., 2024) or labor-intensive long-context data annotation (Zeng et al., 2024). These findings underscore the efficacy of our proposed method in addressing the challenges of long-context alignment while simultaneously preserving short-context capabilities, offering a more efficient and balanced approach to the development of long-context LLMs.

## 2 PRELIMINARIES

In this section, we introduce two key methods for aligning language models with human preferences: Reinforcement Learning from Human Feedback (RLHF, §2.1) and Direct Preference Optimization (DPO, §2.2).

### 2.1 RLHF

Reinforcement Learning from Human Feedback (RLHF) (Ouyang et al., 2022; Stiennon et al., 2020) aims to optimize the policy model $\pi_\theta$ to maximize rewards while maintaining proximity to a reference policy $\pi_{\text{ref}}$. Formally, the objective is

$$\max_{\pi_\theta} \mathbb{E}_{x \sim \mathcal{D}, y \sim \pi_\theta(y|x)} \big[ r_\phi(x, y) \big] - \beta \mathbb{D}_{\text{KL}} \big[ \pi_\theta(y \mid x) \, \| \, \pi_{\text{ref}}(y \mid x) \big], \tag{1}$$

where $r_\phi$ is the reward model that has been trained on ranked responses to reflect human preference, $\beta$ is a hyper-parameter controlling the deviation from reference policy, and $\mathbb{D}_{\text{KL}}$ denotes the Kullback-Leibler divergence. Typically, both $\pi_\theta$ and $\pi_{\text{ref}}$ are initialized with identical model.

### 2.2 DPO

Considering the instability and difficulty of RLHF training, DPO (Rafailov et al., 2023) offers an alternative approach by reparameterizing the reward function $r$ that incorporates the optimal policy:

$$r(x, y) = \beta \log \frac{\pi_\theta(y \mid x)}{\pi_{\text{ref}}(y \mid x)} + \beta \log Z(x), \tag{2}$$

where $Z(x)$ is the partition function. DPO assumes access to preference data $\mathcal{D}$, which consists of paired responses $(y_w, y_l)$ to an instruction $x$. Specifically, the $y_w$ and $y_l$ represent the preferred (winning) and dispreferred (losing) responses, respectively, based on human preference. Inspired by the Bradley-Terry (BT) theory that models the preference distribution $p^*$ by

$$p^*(y_w \succ y_l \mid x) = \sigma(r(x, y_w) - r(x, y_l)), \tag{3}$$

where $\sigma$ is the sigmoid function. DPO derives the preference optimization objective for the policy model $\pi_\theta$ as

$$\mathcal{L}_{\text{DPO}}(\pi_\theta; \pi_{\text{ref}}) = -\mathbb{E}_{(x, y_w, y_l) \sim \mathcal{D}} \left[ \sigma(r_\theta(x, y_w) - r_\theta(x, y_l)) \right]$$
$$= -\mathbb{E}_{(x, y_w, y_l) \sim \mathcal{D}} \left[ \log \sigma \left( \beta \log \frac{\pi_\theta(y_w \mid x)}{\pi_{\text{ref}}(y_w \mid x)} - \beta \log \frac{\pi_\theta(y_l \mid x)}{\pi_{\text{ref}}(y_l \mid x)} \right) \right]. \tag{4}$$

## 3 LONGPO: SHORT-TO-LONG PREFERENCE OPTIMIZATION

Motivated by the challenges of data annotation and performance balance during long-context alignment, we introduce the Short-to-**Long P**reference **O**ptimization (**LongPO**), to effectively empowers a short-context LLM self-evolve to a long-context counterpart while preserving its original short-context capabilities. The foundation of LongPO lies in the transfer of capabilities deeply ingrained during short-context alignment to long-context scenarios by learning from short-to-long preference (§3.1). Additionally, LongPO incorporates a short-to-long constraint based on the KL divergence between short- and long-context models during training, to maintain the short-context performance in a simple yet effective way (§3.2). In §3.3, we present the details of curating short-to-long preference data without external guidance and self-evolving long context training process using LongPO.

### 3.1 LEARNING FROM SHORT-TO-LONG PREFERENCE

As outlined in §2, aligning LLMs with human preference typically relies on datasets comprising ranked responses to identical prompts or instructions. However, in long-context scenarios, constructing such datasets becomes impractical due to the extensive effort required for annotation. To circumvent the external data annotation, we leverage the *short-to-long preference* to internally transfer capabilities well-established in the short-context alignment of LLMs to long-context counterpart.

Concretely, we assume access solely to a short-context LLM $\pi_S$ that has been well aligned. Given a long input $x_L = [C_L; I_L]$ where $C_L$ is the long context and $I_L$ is the instruction, we can acquire the response $y_L \sim \pi_S(y \mid x_L)$ by conditioning on the entire context. Due to the limitations of $\pi_S$ in handling long contexts, $y_L$ is likely to be of lower quality.

We then hypothesize an ideal extractor $\mathcal{F}$ that can rigorously identify and extract all essential information $C_S$ within $C_L$ relevant to addressing $I_L$:

$$C_S = \mathcal{F}(C_L, I_L). \tag{5}$$

By querying the instruction $I_L$ based on $C_S$, we obtain a new answer $y_S \sim \pi_S(y \mid x_S)$, where $x_S = [C_S; I_L]$. As $C_S$ is a shortened context for $I_L$, the well-aligned short-context model $\pi_S$ should be capable of producing a high-quality answer that aligns with human preferences.

Intuitively, $y_S$ can serve as a high-quality answer even when giving the whole long context, as its conditioned context is self-contained for instruction $I_L$. Hence, we definite the short-to-long preference distribution $p^{SL}$ based on Bradley-Terry (BT) model following Eq. (3):

$$p^{SL}(y_S > y_L \mid x_L) = \sigma(r(x_L, y_S) - r(x_L, y_L)). \tag{6}$$

We now steer a policy model $\pi_\theta$ (initialized with $\pi_S$) to follow the preference distribution $p^{SL}$, forming the LongPO objective:

$$\mathcal{L}_{\text{LongPO}}(\pi_\theta; \pi_{\text{ref}}) = -\mathbb{E}_{(x_S, x_L, y_S, y_L) \sim \mathcal{D}^{SL}} \left[ \sigma(r_\theta(x_L, y_S) - r_\theta(x_L, y_L)) \right], \tag{7}$$

where $\mathcal{D}^{SL}$ is the short-to-long preference data consisting of quadruples $(x_S, x_L, y_S, y_L)$. This objective encourages the policy model to consistently accommodate the well-aligned short-context preference while deviating the under-aligned long-context preference. Therefore, LongPO internally transfers preferences from short to long contexts without requiring external supervision, effectively addressing the challenge of long-context data annotation.

### 3.2 SHORT-TO-LONG CONSTRAINT

Long-context alignment often leads to an imbalance between long- and short-context performance. While this issue can be mitigated by carefully calibrating the scale and mixing proportion of long and short data across various context lengths, such an approach is resource-intensive and time-consuming. Moreover, an excessive incorporation of short-context data may inadvertently lead to insufficient long-context alignment. In LongPO, we recognize that the degradation in short-context performance during long-context alignment may be attributed to an improper (or missing) constraint in current alignment methods.

Specifically, the RLHF and DPO objectives (implicitly) include a KL divergence term, $\beta \mathbb{D}_{\text{KL}} \big[ \pi_\theta(y \mid x) \parallel \pi_{\text{ref}}(y \mid x) \big]$, which serves as a constraint to prevent excessive deviation from the reference model in Eq. (1). For a long input $x_L$, this constraint $\mathcal{C}$ is expressed as:

$$\mathcal{C} = \beta \mathbb{D}_{\text{KL}} \big[ \pi_\theta(y \mid x_L) \parallel \pi_{\text{ref}}(y \mid x_L) \big]. \tag{8}$$

However, the reference model is typically the short-context model $\pi_S$ itself, which is not adept at handling long contexts. This results in a problematic reference distribution $\pi_{\text{ref}}(y \mid x_L)$, leading to undesired deviation from the short-context model distribution.

To address this issue, we propose a *short-to-long constraint* leveraging the quadruples introduced in Eq. (7). Recall that $x_S$ contains all the essential information from $x_L$ required to generate a satisfactory response, $\pi_S$ can serve as a proficient reference model conditioned on $x_S$. While for an ideal reference model $\pi_{\text{ref}}^*$ capable of handling context lengths from short to long, we should have:

$$\mathbb{D}_{\text{KL}} \big[ \pi_{\text{ref}}^*(y \mid x_L) \parallel \pi_{\text{ref}}^*(y \mid x_S) \big] = \mathbb{D}_{\text{KL}} \big[ \pi_{\text{ref}}^*(y \mid x_S) \parallel \pi_S(y \mid x_S) \big] = 0, \tag{9}$$

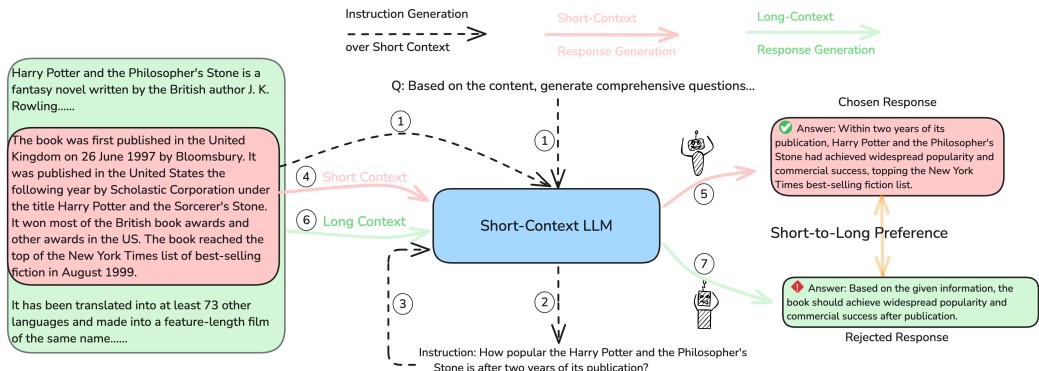

Figure 2: The procedure of generating short-to-long preference data from step 1 to 7.

namely $\pi_{\text{ref}}^*(y \mid x_{\text{L}})$ and $\pi_{\text{S}}(y \mid x_{\text{S}})$ are identical distribution following Gibbs' inequality. We hence derive an adjusted short-to-long constraint between short-context reference model and "long-context" policy model given contexts of different lengths:

$$\mathcal{C}' = \beta \mathbb{D}_{\text{KL}}\big[\pi_\theta(y \mid x_{\text{L}}) \,\|\, \pi_{\text{S}}(y \mid x_{\text{S}})\big]. \tag{10}$$

This refined constraint ensures that the policy model $\pi_\theta$ operating on long contexts does not deviate significantly from the short-context model $\pi_{\text{S}}$ when provided with the essential information. By enforcing this constraint, we aim to preserve the short-context performance during long-context alignment, thereby addressing the imbalance issue in a more principled manner.

By incorporating the short-to-long constraint in Eq. (2), we have a refined reward function for long input $x_{\text{L}}$ (following derivation in Appx. §A.1):

$$r_\theta^{\text{LongPO}}(x_{\text{L}}, y) = \beta \log \frac{\pi_\theta(y \mid x_{\text{L}})}{\pi_{\text{S}}(y \mid x_{\text{S}})} + \beta \log Z(x_{\text{L}}, x_{\text{S}}), \tag{11}$$

where $x_{\text{S}}$ is extracted from $x_{\text{L}}$ as illustrated in Eq. (5). Hence we access the LongPO objective:

$$\begin{aligned}
\mathcal{L}_{\text{LongPO}}(\pi_\theta; \pi_{\text{S}}) &= -\mathbb{E}_{(x_{\text{S}}, x_{\text{L}}, y_{\text{S}}, y_{\text{L}}) \sim \mathcal{D}^{\text{SL}}}\left[\sigma(r_\theta^{\text{LongPO}}(x_{\text{L}}, y_{\text{S}}) - r_\theta^{\text{LongPO}}(x_{\text{L}}, y_{\text{L}}))\right] \\
&= -\mathbb{E}_{(x_{\text{S}}, x_{\text{L}}, y_{\text{S}}, y_{\text{L}}) \sim \mathcal{D}^{\text{SL}}}\left[\log \sigma\left(\beta \log \frac{\pi_\theta(y_{\text{S}} \mid x_{\text{L}})}{\pi_{\text{S}}(y_{\text{S}} \mid x_{\text{S}})} - \beta \log \frac{\pi_\theta(y_{\text{L}} \mid x_{\text{L}})}{\pi_{\text{S}}(y_{\text{L}} \mid x_{\text{S}})}\right)\right].
\end{aligned} \tag{12}$$

### 3.3 SELF-EVOLVING

**Initialization.** LongPO relies solely on access to a well-aligned short-context LLM, i.e., $\pi_{\text{S}}$, in conjunction with a long-context plain corpus. Note that the long-context corpus need not be meticulously crafted, as it can be sampled and extracted from existing pretraining corpora of LLMs.

**Construction of short-to-long preference data.** The construction of short-to-long preference data $\mathcal{D}^{\text{SL}}$ introduced in §3.1 assumes an impractical extractor capable of retrieving essential information from long contexts for each instruction. To satisfy this hypothesis, we reversely prompt $\pi_{\text{S}}$ to generate instructions for shortened chunks within long documents. This ensures that the short context information is self-contained for instructions. Concretely, our data construction process involves two steps as displayed in Figure 2:

1. **Instruction Generation.** For each long document $C_{\text{L}}$, we randomly sample a shortened chunk $C_{\text{S}}$ and prompt the $\pi_{\text{S}}$ to generate an instruction via the Self-Instruct (Wang et al.). To ensure the diversity of instructions, the model is prompted to generate an instruction pool first and then we randomly sample an instruction $I_{\text{L}}$ from this pool.

2. **Response Generation.** Using the generated instruction $I_{\text{L}}$, we prompt $\pi_{\text{S}}$ to produce two responses: a chosen response $y_{\text{S}} \sim \pi_{\text{S}}(y \mid x_{\text{S}})$ based on the short context $x_{\text{S}}$, and a rejected response $y_{\text{L}} \sim \pi_{\text{S}}(y \mid x_{\text{L}})$ derived from the long context $x_{\text{L}}$.

**Iterative self-evolving training with LongPO.** LongPO employs an iterative process to extend LLM context length. Initially, a short-context LLM $\pi_S$ generates short-to-long preference data for documents of length $L^1$. The resulting model after LongPO training, now capable of handling $L^1$ contexts, then serves as the new "short-context LLM" for the next iteration, generating data for an extended length $L^2$. This process repeats, progressively increasing context length capacity.

As there are multiple short chunks $C_S = \{C_S^i\}_{i=1}^n$ within a long document $C_L$, we collect the instruction-response triples $(I_L^i, y_S^i, y_L^i)$ for each chunk within identical long document, to form a multi-turn dataset $\hat{\mathcal{D}}^{SL}$. We then aggregate the probabilities across all turns to produce a multi-turn LongPO objective:

$$\mathcal{L}_{LongPO}^{MT}(\pi_\theta; \pi_S) = -\mathbb{E}_{(x_S, x_L, y_S, y_L) \sim \hat{\mathcal{D}}^{SL}} \left[ \log \sigma \left( \beta \log \frac{\sum_{i=1}^n \pi_\theta(y_S^i \mid x_L^i)}{\sum_{i=1}^n \pi_S(y_S^i \mid C_S^i)} - \beta \log \frac{\sum_{i=1}^n \pi_\theta(y_L^i \mid x_L^i)}{\sum_{i=1}^n \pi_S(y_L^i \mid C_S^i)} \right) \right],$$
(13)

where $x_S = \{[C_S^i; I_L^i]\}_{i=1}^n$, $x_L = \{[C_L; I_L^i]\}_{i=1}^n$, $y_S = \{y_S^i\}_{i=1}^n$, and $y_L = \{y_L^i\}_{i=1}^n$. LLMs trained with LongPO do not necessarily involve continual training before, which may lead to instability when processing long contexts. To address this issue and stabilize the training process, we incorporate a continual training objective following Pang et al. (2024b). Specifically, we add the negative log-likelihood (NLL) loss over entire long chosen sequences $S_L = [x_L; \{I_L^i; y_S^i\}_{i=1}^n]$ to LongPO objective. Thus, our final training objective is:

$$\mathcal{L}_\theta = \lambda \cdot \mathcal{L}_{LongPO}^{MT}(\pi_\theta; \pi_S) + \mathcal{L}_{NLL}(\pi_\theta; S_L) = \lambda \cdot \mathcal{L}_{LongPO}^{MT}(\pi_\theta; \pi_S) + \frac{\pi_\theta(S_L)}{|S_L|}.$$
(14)

# 4 EXPERIMENTAL SETUP

## 4.1 TRAINING SETUP

**Data Curation Details.** We curate the short-to-long preference data based on a long-context corpus sampled from the Book and ArXiv subsets of Long-Data-Collection[1], and the GitHub subset of RedPajama (Computer, 2023). For a specific target length (e.g., 128K tokens), we filter the corpus to include only documents that are shorter than this length but longer than 64K tokens. Each long document is then segmented into chunks of up to 32K tokens, with a maximum of 4 randomly-sampled chunks retained per document. For instruction generation, we prompt short-context models to generate 4 instructions per document, from which we randomly select one for further use. After filtering undesired (censored and repetitive) responses, we collect 45K, 16K, 2.5K multi-turn instruction-response samples of 128K, 256K, and 512K tokens for Mistral-7B, and 32K samples of 128K tokens for Qwen2.5-7B. More details are listed in Appx. §B.1.

**Training Details.** We extend the context length of Mistral-7B and Qwen2.5-7B using our LongPO on short-to-long preference data specifically generated by models themselves. The training process begins with utilizing Mistral-7B/Qwen2.5-7B to generate data with a length of 128K and extend the context length to 128K. To investigate the scalability, we utilize the resulting Mistral-7B-LongPO-128K to generate data with lengths of 256K/512K and further extend the context length to 256K/512K. We leverage Deepspeed-Ulysses (Jacobs et al., 2023) for sequence parallelism and employ Flash Attention (Dao et al., 2022; Dao, 2023) for efficient computation. All models are optimized using the Adam optimizer (Kingma & Ba, 2015) with a learning rate of 5e-7. We set the margin $\beta$ in Eq. (13) to 0.1 and the weighting factor $\lambda$ in Eq. (14) to 0.01. RoPE $\theta$ is set as 1e7 and the batch size is set as 8. More details are listed in Appx. §B.3.

## 4.2 EVALUATION BENCHMARKS

We assess both the long- and short-context capabilities of our models against baselines. The long-context evaluation utilizes the following benchmarks:

- ∞**Bench** (Zhang et al.). We evaluate all models on three tasks in this benchmark: summarization (En.Sum), long-book question answering (En.QA), and multi-choice question-answering (En.MC). The evaluation length is beyond 100K.

[1]https://huggingface.co/datasets/togethercomputer/Long-Data-Collections

- **RULER** (Hsieh et al., 2024). This benchmark comprises four types of synthetic tasks across variable sequence lengths (4K to 128K): Needle-in-a-haystack (NIAH) retrieval, Multi-hop Tracing with Variable Tracking (VT), Aggregation, and Question Answering (QA). We exclude the Aggregation tasks, which involve word frequency counting within the context, since they present challenges in word counting beyond mere long-context capabilities that current LLMs still struggle in.
- **LongBench-Chat** (Bai et al., 2024). This benchmark assesses instruction-following abilities over long contexts (10K to 100K tokens), employing GPT-4-128K as an impartial judge to evaluate model-generated responses. We filter out the English samples for fair comparison across different models.

For short-context evaluation, we employ MMLU (Hendrycks et al., 2021), ARC-C (Clark et al., 2018), Hellaswag (Zellers et al., 2019) and Winogrande (Sakaguchi et al., 2019) for assessing the general language understanding and reasoning capabilities, and MT-Bench (Zheng et al., 2023) for assessing instruction-following capability. More details are listed in Appx. §B.2.

## 4.3 BASELINES

We train our LongPO on Mistral-7B-Instruct-v0.2 (denoted as Mistral-7B) and Qwen2.5-7B-Instruct (denoted as Qwen2.5-7B), comparing them against a range of powerful LLMs including GPT-4-128K, Qwen2-72B-Instruct (Yang et al., 2024), LLaMA-3.1-70B, LLaMA-3.1-8B, GLM-4-9B-Chat, GLM-4-9B-Chat-1M, LWM-Text-Chat-1M (Liu et al., 2024b), and Yarn-Mistral-7b-128k (Peng et al., 2023). Additionally, we establish baselines using Mistral-7B trained with conventional SFT and DPO on the same dataset used by LongPO.

For short-context evaluation, we primarily compare the performance of naive LLMs against their counterparts post-trained with SFT, DPO, and LongPO on our synthetic data. To provide a more comprehensive comparison, we also include two series of open-source long-context language models: GLM-4-9B-Chat versus GLM-4-9B-Chat-1M, and LWM-Text-Chat-128k versus LWM-Text-Chat-1M. This allows us to assess the effectiveness of our LongPO to maintain the short-context performance during long-context alignment, comparing with baselines utilizing various strategies.

## 5 RESULTS AND ANALYSES

In this section, we demonstrate the exceptional effectiveness of LongPO through two types of comparisons: (1) comparison with naive SFT and DPO trained on identical models and datasets; (2) comparison with SOTA long-context LLMs.

### 5.1 COMPARISON WITH SFT AND DPO

We first compare LongPO with conventional SFT and DPO using identical LLM (Mistral-7B). All models are trained on equivalent self-generated datasets, as detailed in §4.1. Given the inability of SFT to leverage preference data, we apply it to the instructions paired with chosen responses.

**LongPO exhibits superior performance over SFT and DPO.** The experimental results, illustrated in Table 1, reveal consistent and substantial performance gains (10 to 20+ points) of LongPO over SFT and DPO across a diverse range of long-context tasks. Crucially, as depicted in Figure 3, LongPO maintains robust short-context performance compared with original short-context LLMs (59.99 vs 59.15 on MMLU), whereas SFT and DPO exhibit notable degradation in short-context scenarios after long-context alignment process.

The performance disparity between LongPO and SFT can be attributed to the explicit integration of short-to-long preference in LongPO, which is either absent or merely implicit in the chosen responses utilized by SFT. While both LongPO and DPO leverage the proposed short-to-long preference data, the pivotal difference lies in the short-to-long constraint introduced in §3.2. The marked performance gaps between LongPO and DPO, observed across both long- and short-context tasks, highlight the effectiveness of the proposed constraint for successfully mitigating the problematic limitations in DPO and retaining the short-context performance during long-context training. More ablations are detailed in §5.3.

Table 1: Long-Context Performance of our LongPO compared with baselines. Higher is better for all metrics. Results marked with ♭ are evaluated by ourselves, while other results of baselines are sourced from the original benchmarks. Full results on RULER are listed in Table 2.

| Model | Train/Claimed Length | ∞Bench | | | | RULER | | | | LongBench-Chat (EN) |
|---|---|---|---|---|---|---|---|---|---|---|
| | | En.Sum | En.QA | En.MC | AVG. | NIAH | VT | QA | AVG. | |
| GPT-4-128K | 128K | 14.73 | 22.44 | 67.25 | 34.81 | 95.4 | 99.9 | 70.3 | 88.53 | 8.40 |
| Qwen2-72B | 128K | 24.32♭ | 7.03♭ | 72.05♭ | 34.47♭ | 88.6 | 95.7 | 66.7 | 83.67 | 7.72♭ |
| LLaMA 3.1-70B | 128K | 33.55♭ | 36.08♭ | 69.00♭ | 46.21♭ | 96.1 | 93.2 | 67.8 | 85.7 | 6.67♭ |
| LLaMA 3.1-8B | 128K | 28.06♭ | 30.47♭ | 58.08♭ | 38.87♭ | 97.93 | 91.4 | 64.7 | 84.68 | 6.22♭ |
| GLM-4-9B | 128K | 14.84♭ | 9.51♭ | 67.25♭ | 30.53♭ | 96.51♭ | 97.3♭ | 64.8♭0 | 86.20♭ | 5.67♭ |
| GLM-4-9B-1M | 1M | 28.3 | 9.7 | 68.6 | 35.53 | 98.2 | 99.4 | 69.4 | 89.0 | 5.03♭ |
| LWM-7B-1M | 1M | 4.33♭ | 0.0♭ | 3.06♭ | 2.46♭ | 87.20 | 57.5 | 56.4 | 67.03 | 1.25♭ |
| YaRN-Mistral-7B | 128K | 9.09 | 9.55 | 27.95 | 15.53 | 63.4 | 36.1 | 25.9 | 41.8 | - |
| Mistral-7B | 32K | 22.13 | 4.93 | 14.41 | 13.82 | 72.60 | 74.40 | 52.2 | 66.4 | 4.10 |
| - SFT | 128K | 23.44 | 13.45 | 53.21 | 30.03 | 88.73 | 79.64 | 51.08 | 73.15 | 4.25 |
| - DPO | 128K | 15.21 | 10.34 | 48.14 | 25.56 | 74.25 | 72.36 | 50.24 | 65.62 | 4.08 |
| - LongPO (iter1) | 128K | 27.05 | 23.51 | 67.25 | 39.27 | 96.88 | 96.49 | 64.81 | 86.06 | 5.42 |
| - LongPO (iter2) | 256K | 28.16 | 24.43 | 66.35 | 39.65 | 96.80 | 97.0 | 64.87 | 86.22 | 5.48 |
| - LongPO (iter3) | 512K | 29.10 | 27.85 | 66.67 | 41.21 | 97.28 | 97.48 | 64.92 | 86.56 | 5.80 |
| Qwen2.5-7B | 128K | 22.89 | 6.08 | 52.4 | 27.12 | 82.1 | 80.09 | 54.30 | 72.16 | 5.80 |
| - LongPO (iter1) | 128K | 32.06 | 17.32 | 72.05 | 40.48 | 95.81 | 89.71 | 59.4 | 81.64 | 5.75 |

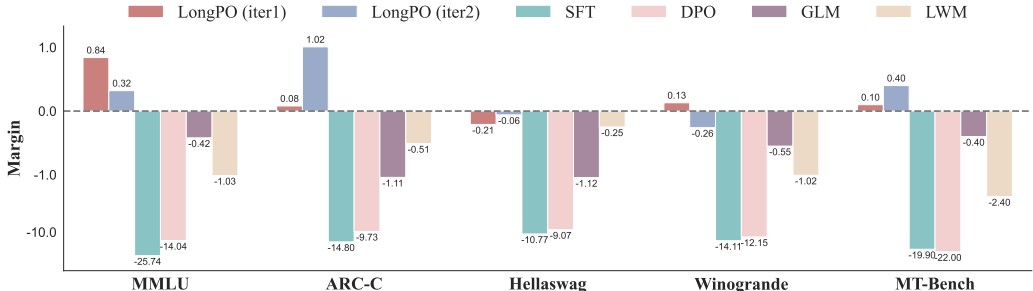

Figure 3: The margins of the short-context performance of Mistral-7B-LongPO and baselines relative to corresponding base model. GLM and LWM refer to the margins of GLM-9B-1M and LWM-7B-1M over GLM-9B-128K and LWM-7B-128K, respectively. MT-Bench metrics (∈[0, 10]) are linearly scaled to [0, 100] for better comparability across tasks. See numerical results in Table 3.

## 5.2 COMPARISON WITH SOTA LONG-CONTEXT LLMS

To further substantiate the efficacy of LongPO, we conducted an extensive comparison between our LongPO-trained Mistral-7B and leading long-context LLMs across varying model scales.

**LongPO demonstrates exceptional competitiveness at similar scale.** As detailed in Table 1, LongPO demonstrates formidable competitiveness in terms of models at similar scales. For example, Mistral-7B-LongPO significantly outperforms some established long-context models, including LWM-7B and YaRN-Mistral, across all long-context tasks in ∞Bench and RULER. Remarkably, Mistral-7B-LongPO-128K surpasses GLM-4-9B (39.27 vs. 30.53 on ∞Bench and 86.06 vs. 86.20 on RULER), although the latter is training on manually annotated long-context data spanning up to 128K sequence length. Moreover, GLM-4-9B-1M, an extension of GLM-4-9B trained on contexts up to 1M tokens, demonstrates slightly superior performance than LongPO on the RULER benchmark. However, these performance gains come at the costs of *degenerated short-context performance* (0.41 on MMLU) and *long-context instruction-following capability* (0.64 on LongBench-Chat (EN)) as illustrated in Figure 3. Notably, our models still outperform GLM-4-9B-1M on ∞Bench even trained with substantially shorter sequences. These results underscore the exceptional efficiency of LongPO in transferring performance from short to long contexts through self-evolution, thereby circumventing the need for extensive manual annotation.

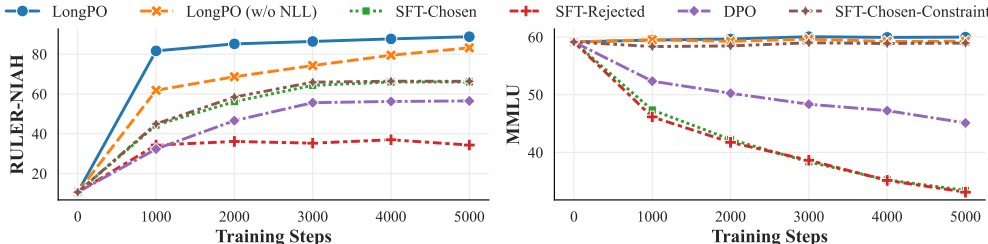

Figure 4: Long- and short-context performance comparison among LongPO, SFT on chosen responses (**SFT-Chosen**), SFT on rejected responses (**SFT-Rejected**), DPO, and SFT on chosen responses with short-to-long constraint (**SFT-Chosen-Constraint**).

**Long-context annotation is not sufficient.**   The superiority of our approach is particularly evident in the En.QA task within ∞Bench, which involves complex free-form question answering over extensive book-length contexts. In this challenging task, our models surpass both GLM-4-9B and GLM-4-9B-1M by substantial margins (10+ points). The inherent difficulty of such task, which poses challenges even for human annotators, highlights the limitations of relying solely on manually annotated long-context data. By effectively transferring short-context capabilities to long-context scenarios, LongPO demonstrates superior scalability and efficacy across diverse and intricate tasks.

**Superior LLMs Yet to Dominate Long-Context Scenarios**   When benchmarked against leading models such as GPT-4-128K, our LongPO-trained models still exhibit comparable or even superior long-context performance (e.g., Mistral-7B-LongPO-128K of 39.27 vs. GPT-4-128K of 34.81 on ∞Bench), despite being based on significantly smaller Mistral-7B. This observation reveals that even the most advanced LLMs have not yet achieved the same level of dominance in long-context scenarios as they have in short-context tasks. This performance gap can be attributed primarily to the scarcity of high-quality, large-scale long-context training data. The dearth of such data is particularly impactful for larger LLMs, given the established scaling laws in language model training. This finding underscores the potential of LongPO for enhanced performance without the requirement for externally annotated long-context datasets.

## 5.3   ABLATION STUDIES

We conduct comprehensive ablation studies to investigate the efficacy of components in LongPO:

**Effectiveness of short-to-long preference.**   The core of LongPO is learning the short-to-long preference between chosen and rejected responses given short and long contexts, respectively. To evaluate this component's effectiveness, we compare LongPO with two baseline methods: SFT on chosen responses (SFT-Chosen) and on rejected responses (SFT-Rejected). SFT-Chosen implicitly incorporates short-context preference, while SFT-Rejected entirely omits it. As illustrated in Figure 4, LongPO consistently outperforms both SFT variants in long-context performance (RULER-NIAH) throughout the training process. This substantial improvement underscores the efficacy of our short-to-long preference approach in enhancing long-context capabilities.

**Effectiveness of short-to-long constraint.**   To assess the impact of our short-to-long constraint, we compare LongPO with DPO upon short-to-long preference that removes this constraint. As evident in Figure 4, the unconstrained DPO demonstrates markedly inferior performance throughout the training process, both in long- and short-context tasks. Notably, short-context capabilities degrade rapidly in DPO during the initial training. Conversely, when we apply our short-to-long constraint to naive SFT without explicit short-to-long preference, the model maintains short-context performance on par with the original LLMs, even after long-context alignment. These results demonstrate the crucial role of our short-to-long constraint in preserving short-context capabilities.

**Impact of NLL loss.**   We investigate the effect of incorporating a negative log-likelihood (NLL) loss over long context input and chosen response in Eq. (14) during LongPO training. As shown in Figure 4, removing the NLL loss significantly degrades the long-context performance of LongPO

across the training procedure. Specifically, the convergence of training for long-context performance becomes slower. This demonstrates the crucial role of NLL loss in enhancing long-context capabilities without resorting to continual training on long data.

## 6 RELATED WORK

**Alignment of LLMs.** Aligning Large Language Models (LLMs) with human preferences and values has been crucial to unlocking their full potential from large-scale pretraining. The typical alignment process begins with Supervised Finetuning (SFT) on annotated instruction-response pairs. This is followed by Reinforcement Learning from Human Feedback (RLHF), which aligns LLMs more closely with human intentions through reward model training and policy optimization (Christiano et al., 2017; Ouyang et al., 2022; Bai et al., 2022; Stiennon et al., 2020). To streamline RLHF training, Direct Preference Optimization (DPO) (Rafailov et al., 2023) and its variants (Ethayarajh et al., 2024; Azar et al., 2023; Pang et al., 2024a; Hong et al., 2024; Meng et al., 2024) have been proposed, eliminating the need for explicit reward model training by learning preferences directly from human-ranked response pairs. While these alignment methods have shown significant success, they heavily rely on human-annotated data. This reliance becomes problematic for long-context data, where human annotation is both challenging and potentially less reliable.

**Long-context extending of LLMs.** Extending the context length of LLMs has been approached through various methods. Some techniques involve scaling the rotary position embedding (Su et al., 2022) followed by continual training on a small corpus of long documents (Chen et al., 2023b; Peng et al., 2023; Rozière et al., 2023; Chen et al., 2023a). Alternative approaches, such as those proposed by Jin et al. (2024); An et al. (2024), introduce hierarchical or chunked attention mechanisms to extend context length without additional training. However, these methods often involve limitations in practical applications. Recent advancements include the work of Dubey et al. (2024), who proposed continual pretraining on a massive long-context corpus (800B tokens) and incorporating a small fraction (0.1%) of long-context data during SFT to enhance long-context capabilities. Zeng et al. (2024) utilizes human-annotated long-context data for SFT and DPO to align long-context LLMs. Despite their effectiveness, these methods require either extensive training or human annotation of long-context data, making them prohibitively expensive and lack scalability.

**Self-Evolving LLMs.** Recent works (Yuan et al., 2024; Liu et al., 2024a; Li et al., 2024) have unveiled the remarkable capability of Large Language Models (LLMs) to evolve from relatively weak to significantly stronger performance through self-augmented data. Yuan et al. (2024); Liu et al. (2024a) leverage iterative training on model-generated responses, ranked by LLM-as-a-Judge (Zheng et al., 2023) prompting, to enhance model itself. Li et al. (2024) introduces the instruction backtranslation to produce self-augmenting data that further enhances model capabilities. Our work first extends the self-evolution property to the context length, to develop long-context LLMs without relying on external annotations.

## 7 CONCLUSION AND DISCUSSION

In this work, we propose LongPO, a novel long-context alignment method that enables LLMs to effectively transfer their short-context capabilities to long-context scenarios. Our approach addresses key challenges in long-context alignment by leveraging intrinsic model knowledge, eliminating the need for external long-context annotated data. LongPO is built on short-to-long preference data, comprising paired responses for the same instruction given a long context and relevant shortened chunk, respectively. By steering the policy model to learn from the discrepancies within these paired responses, LongPO facilitates the transfer of established capabilities from short to long contexts. In addition, LongPO incorporates a short-to-long constraint using KL divergence, that effectively preserve short-context performance during training. Experimental results demonstrate that LongPO significantly improves long-context performance across various tasks, outperforming existing alignment methods and even surpassing more sophisticated models. Importantly, this improvement is achieved without sacrificing short-context proficiency. The success of LongPO highlights the potential of leveraging internal model knowledge for alignment tasks, opening new avenues for efficient adaptation of LLMs to diverse context lengths.

ACKNOWLEDGMENTS

This work was supported by DAMO Academy through DAMO Academy Research Intern Program.

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

## A MATHEMATICAL DERIVATIONS

### A.1 DERIVING THE LONGPO OBJECTIVE

In this section, we will derive the reward function of our LongPO objective in Eq. (11) by incorporating short-to-long constraint in Eq. (10). Starting from RLHF objective in Eq. (1) with short-to-long constraint, we have

$$\max_{\pi} \mathbb{E}_{x_{\mathrm{L}} \sim \mathcal{D}, y \sim \pi}\big[r(x_{\mathrm{L}}, y)\big] - \beta \mathbb{D}_{\mathrm{KL}}\big[\pi(y|x_{\mathrm{L}}) \| \pi_{\mathrm{S}}(y|x_{\mathrm{S}})\big] \tag{15}$$

Following the DPO derivation process (Rafailov et al., 2023), we have:

$$\max_{\pi} \mathbb{E}_{(x_{\mathrm{L}}, x_{\mathrm{S}}) \sim \hat{\mathcal{D}}^{\mathrm{SL}}, y \sim \pi(y|x_{\mathrm{L}})}\big[r(x_{\mathrm{L}}, y)\big] - \beta \mathbb{D}_{\mathrm{KL}}\big[\pi(y|x_{\mathrm{L}}) \| \pi_{\mathrm{S}}(y|x_{\mathrm{S}})\big]$$

$$= \max_{\pi} \mathbb{E}_{(x_{\mathrm{L}}, x_{\mathrm{S}}) \sim \hat{\mathcal{D}}^{\mathrm{SL}}} \mathbb{E}_{y \sim \pi(y|x_{\mathrm{L}})}\left[r(x_{\mathrm{L}}, y) - \beta \log \frac{\pi(y|x_{\mathrm{L}})}{\pi_{\mathrm{S}}(y|x_{\mathrm{S}})}\right]$$

$$= \min_{\pi} \mathbb{E}_{(x_{\mathrm{L}}, x_{\mathrm{S}}) \sim \hat{\mathcal{D}}^{\mathrm{SL}}} \mathbb{E}_{y \sim \pi(y|x_{\mathrm{L}})}\left[\log \frac{\pi(y|x_{\mathrm{L}})}{\pi_{\mathrm{S}}(y|x_{\mathrm{S}})} - \frac{1}{\beta} r(x_{\mathrm{L}}, y)\right]$$

$$= \min_{\pi} \mathbb{E}_{(x_{\mathrm{L}}, x_{\mathrm{S}}) \sim \hat{\mathcal{D}}^{\mathrm{SL}}} \mathbb{E}_{y \sim \pi(y|x_{\mathrm{L}})}\left[\log \frac{\pi(y|x_{\mathrm{L}})}{\frac{1}{Z(x_{\mathrm{L}}, x_{\mathrm{S}})}\pi_{\mathrm{S}}(y|x_{\mathrm{S}}) \exp\left(\frac{1}{\beta} r(x_{\mathrm{L}}, y)\right)} - \log Z(x_{\mathrm{L}}, x_{\mathrm{S}})\right], \tag{16}$$

where we have partition function:

$$Z(x_{\mathrm{L}}, x_{\mathrm{S}}) = \sum_{y} \pi_{\mathrm{S}}(y|x_{\mathrm{S}}) \exp\left(\frac{1}{\beta} r(x_{\mathrm{L}}, y)\right).$$

The partition function is only related to $x_{\mathrm{L}}$, $x_{\mathrm{S}}$, and original short-context LLM $\pi_{\mathrm{S}}$. Hence we have the optimal solution following Rafailov et al. (2023):

$$\pi^*(y|x_{\mathrm{L}}) = \frac{1}{Z(x_{\mathrm{L}}, x_{\mathrm{S}})} \pi_{\mathrm{S}}(y|x_{\mathrm{S}}) \exp\left(\frac{1}{\beta} r(x_{\mathrm{L}}, y)\right). \tag{17}$$

The optimal reward function would be derived:

$$r^*(x_{\mathrm{L}}, y) = \beta \log \frac{\pi^*(y|x_{\mathrm{L}})}{\pi_{\mathrm{S}}(y|x_{\mathrm{S}})} + \beta \log Z(x_{\mathrm{L}}, x_{\mathrm{S}}). \tag{18}$$

We thus have:

$$p^*(y_1 > y_2|x_{\mathrm{L}}) = \frac{\exp\left(r^*(x_{\mathrm{L}}, y_1)\right)}{\exp\left(r^*(x_{\mathrm{L}}, y_1)\right) + \exp\left(r^*(x_{\mathrm{L}}, y_2)\right)}$$

$$= \frac{\exp\left(\beta \log \frac{\pi^*(y_1|x_{\mathrm{L}})}{\pi_{\mathrm{S}}(y_1|x_{\mathrm{S}})} + \beta \log Z(x_{\mathrm{L}}, x_{\mathrm{S}})\right)}{\exp\left(\beta \log \frac{\pi^*(y_1|x_{\mathrm{L}})}{\pi_{\mathrm{S}}(y_1|x_{\mathrm{S}})} + \beta \log Z(x_{\mathrm{L}}, x_{\mathrm{S}})\right) + \exp\left(\beta \log \frac{\pi^*(y_2|x_{\mathrm{L}})}{\pi_{\mathrm{S}}(y_2|x_{\mathrm{S}})} + \beta \log Z(x_{\mathrm{L}}, x_{\mathrm{S}})\right)}$$

$$= \frac{1}{1 + \exp\left(\beta \log \frac{\pi^*(y_2|x_{\mathrm{L}})}{\pi_{\mathrm{S}}(y_2|x_{\mathrm{S}})} - \beta \log \frac{\pi^*(y_1|x_{\mathrm{L}})}{\pi_{\mathrm{S}}(y_1|x_{\mathrm{S}})}\right)}$$

$$= \sigma\left(\beta \log \frac{\pi^*(y_1|x_{\mathrm{L}})}{\pi_{\mathrm{S}}(y_1|x_{\mathrm{S}})} - \beta \log \frac{\pi^*(y_2|x_{\mathrm{L}})}{\pi_{\mathrm{S}}(y_2|x_{\mathrm{S}})}\right).$$

By optimizing the $\pi_\theta$ towards the optimal policy $\pi^*$, we finally access the objective of LongPO in Eq. (12).

## B EXPERIMENTAL DETAILS

### B.1 DATA CONSTRUCTION DETAILS

We prompt the Mistral-7B-Instruct-v0.2 to generate instructions with decode parameters of temperature $T = 0.7$ and $p = 0.9$. The prompt of Self-Instruct to generate an instruction pool is shown

Based on the content presented above, generate 4 comprehensive English questions that test a reader's comprehension, analytical skills, and ability to extract and interconnect key themes and ideas across the entire document.

Each question should:
1. Encourage the reader to draw connections between different sections or concepts within the text.
2. Challenge the reader to not only recall information but also to synthesize and summarize the material in a coherent manner.
3. Be unique in its focus, avoiding repetition and ensuring a broad coverage of the document's content.
4. Stimulate critical thinking by requiring the application or evaluation of the text's information in broader contexts or hypothetical scenarios, if relevant.
5. Ensure the question is clear and unambiguous.

Please directly give the questions without verbose illustration, and format the 4 questions numerically from "1:" to "4:".

Figure 5: The prompt for generating instruction pool.

in Figure 5. For generating the corresponding responses, we directly concatenate the short or long context with corresponding instructions and adopt the greedy decoding to maintain the deterministic behaviour of LLMs. As shown in Figure 6, the model would tend to prefer the high-quality chosen response and deviate from the low-quality rejected response over long context, hence improve the long-context capabilities.

## B.2 EVALUATION DETAILS

On long-context benchmarks InfiniteBench and RULER, we evaluate our models and all baselines following the settings in the original benchmarks. For short-context evaluation, we utilize the lm-evaluaton-harness framework (Gao et al., 2024) and following the evaluation settings in (Beeching et al., 2023): 5-shots for MMLU, 25-shots for ARC-C, 10-shots for Hellaswag, and 5-shots for Winogrande. We use GPT-4-Turbo-1106-Preview as the judge for MT-Bench and LongBench-Chat evaluation.

## B.3 MORE TRAINING DETAILS

Leveraging the DeepSpeed-Ulysses sequence parallel framework, we train the Mistral-7B/Qwen2.5-7B with a sequence length of 128K on an 8×A800 80GB, achieving a throughput of 4,401 tokens per second. For sequence lengths of 256K and 512K, the models are trained on a 16×A800 80GB, yielding throughputs of 4,120 tokens per second and 2,744 tokens per second, respectively. To facilitate a comparison with standard LLM alignment methods, we train Mistral-7B using SFT and DPO utilizing the same short-to-long preference data of LongPO. For DPO training, we apply the same settings as LongPO outlined in §4.1, but excluding the short-to-long constraint of LongPO introduced in §3.2. Since SFT cannot utilize paired responses within preference data, we train it using only the chosen responses provided alongside long context inputs. The hyperparameters for SFT remain unchanged, except for an increase in the learning rate to 2e-5.

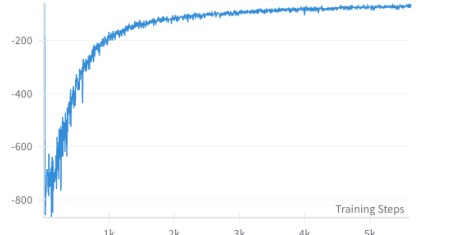 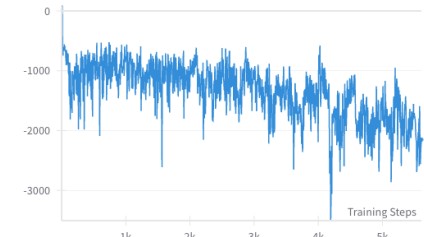

(a) The rewards for chosen response during training.    (b) The rewards for rejected response during training.

Figure 6: The chosen and rejected rewards during the training of Mistral-7B-LongPO-128K.

Table 2: Full results on 13 tasks of RULER benchmark. The **bold** values denote the average score of 13 tasks in RULER over various context lengths.

| Model | Category | 4k | 8k | 16k | 32k | 64k | 128k | AVG |
|---|---|---|---|---|---|---|---|---|
| | NIAH | 99.69 | 98.45 | 97.82 | 95.24 | 74.56 | 26.86 | 82.10 |
| | VT | 99.88 | 99.72 | 96.24 | 96.44 | 81.44 | 6.84 | 80.09 |
| Qwen2.5-7B-Instruct | AGG | 92.52 | 89.78 | 92.08 | 81.93 | 62.48 | 28.23 | 74.50 |
| | QA | 71.00 | 65.30 | 64.00 | 58.70 | 46.80 | 19.99 | 54.30 |
| | AVG (13 tasks) | 94.19 | 92.11 | 91.61 | 87.66 | 68.96 | 24.47 | **76.50** |
| | NIAH | 99.64 | 98.97 | 97.80 | 95.54 | 94.80 | 88.15 | 95.82 |
| | VT | 99.96 | 99.92 | 96.12 | 86.24 | 78.20 | 77.80 | 89.71 |
| Qwen2.5-7B-LongPO-128K | AGG | 95.50 | 86.12 | 91.75 | 82.56 | 66.31 | 49.81 | 78.67 |
| | QA | 70.00 | 64.00 | 62.70 | 57.70 | 53.00 | 49.00 | 59.40 |
| | AVG (13 tasks) | 94.47 | 91.69 | 91.34 | 87.00 | 82.71 | 75.43 | **87.11** |
| | NIAH | 99.43 | 98.64 | 98.09 | 97.84 | 95.82 | 91.44 | 96.88 |
| | VT | 99.40 | 99.16 | 98.08 | 96.36 | 92.80 | 93.12 | 96.49 |
| Mistral-7B-LongPO-128K | AGG | 88.31 | 82.91 | 92.23 | 72.775 | 46.305 | 46.79 | 71.55 |
| | QA | 71.10 | 70.15 | 66.60 | 65.80 | 61.00 | 54.20 | 64.81 |
| | AVG (13 tasks) | 93.36 | 91.88 | 92.35 | 88.94 | 82.61 | 78.97 | **88.02** |
| | NIAH | 99.16 | 97.79 | 98.02 | 97.76 | 96.53 | 91.54 | 96.80 |
| | VT | 99.40 | 99.20 | 97.96 | 97.72 | 94.21 | 93.52 | 97.00 |
| Mistral-7B-LongPO-256K | AGG | 87.40 | 76.59 | 89.03 | 72.20 | 45.17 | 44.47 | 69.14 |
| | QA | 71.50 | 69.50 | 66.70 | 64.30 | 60.80 | 56.40 | 64.87 |
| | AVG (13 tasks) | 93.11 | 90.28 | 91.81 | 88.68 | 82.95 | 79.04 | **87.65** |
| | NIAH | 99.19 | 97.78 | 98.06 | 97.69 | 96.62 | 94.36 | 97.28 |
| | VT | 99.44 | 99.16 | 98.04 | 97.80 | 95.92 | 94.52 | 97.48 |
| Mistral-7B-LongPO-512K | AGG | 87.56 | 76.71 | 88.95 | 72.70 | 44.93 | 44.51 | 69.22 |
| | QA | 71.40 | 69.50 | 66.40 | 64.50 | 60.60 | 57.10 | 64.92 |
| | AVG (13 tasks) | 93.14 | 90.29 | 91.78 | 88.75 | 83.07 | 80.97 | **88.00** |

Table 3: Performance on short-context tasks.

| Model | MMLU | ARC-C | Hellaswag | Winogrande | MT-Bench |
|---|---|---|---|---|---|
| Mistral-7B-Instruct-v0.2 | 59.15 | 59.26 | 83.2 | 78.4 | 6.34 |
| Mistral-7B-LongPO-128K | 59.99 | 59.34 | 82.99 | 78.53 | 6.35 |
| Mistral-7B-LongPO-256K | 59.47 | 60.28 | 83.14 | 78.14 | 6.38 |
| Mistral-7B-LongPO-512K | 59.51 | 60.58 | 82.87 | 77.66 | 6.34 |
| Qwen2.5-7B-Instruct | 74.28 | 67.15 | 81.41 | 74.66 | 7.30 |
| Qwen2.5-7B-LongPO-128K | 73.64 | 65.70 | 80.82 | 74.98 | 7.62 |

