# OpenReview forum: "LongPO: Long Context Self-Evolution of Large Language Models through Short-to-Long Preference Optimization"
_ICLR.cc/2025/Conference — ICLR 2025 Poster_

### Official Review · Reviewer_6PbM · 2024-10-24

**Soundness:** 2
**Presentation:** 3
**Contribution:** 2
**Rating:** 6
**Confidence:** 3

**Summary:**

This paper primarily talks about ways to reinforce the long-context capabilities of LLMs and proposes a novel method, called LongPO, that enables LLMs, which is solely good at short-context tasks in the beginning, to excel on long-context tasks by learning from self-generated short-to-long preference data. LongPO demonstrates promising long-context alignment performance, while retaining the effectiveness of dealing with short-context tasks. Furthermore, this work shows that large-scale human annotation for extended contexts is no longer necessary to excel the model’s long-context capabilities, which sheds light on doing long-context alignment solely with synthetic data.

**Strengths:**

- The idea of constructing short-to-long preference data is simple, straightforward, and easy to implement.
- The performance achieved by the proposed method is impressive compared to the baselines used in the paper, even surpassing some well-known proprietary LLMs.
- The proposed method, LongPO, is able to retain the short-context performance by adding a KL constraint term to the objective, which explicitly guides the model to minimize the deviation from its short-context output distribution.
- The paper is well written with related works being presented thoroughly to highlight its significant contribution.

**Weaknesses:**

- Though the construction process of preference data is pretty clear to me, I’m not fully convinced by the motivation that learning from these paired preference data, in which the longer one is dispreferred, enables the model to excel on long-context capabilities. And the authors haven’t provided good intuition or desired empirical evidence showing why doing preference optimization, where the dispreferred data contains the desired properties, is better than proper baselines, e.g., SFT with high-quality long-context data (in the paper, the SFT baseline is using either chosen short-context data or long-context data, in which the instruction is generated by short-context model).
- The paper doesn’t discuss the bottleneck/limitation of the proposed method. Is LongPO able to scale to much longer contexts, such as 512k, 1 million? When does this method plateaus the performance?

**Questions:**

Do you think some other ways of constructing preference data using simple and clever rules/metrics, eliminating the need for human annotation but ensuring one of chosen or rejected data has a long context, are able to achieve similar alignment performance as LongPO? For example, thinking of constructing a new preference dataset by sampling some short-context data from the chosen group of existing preference datasets as the new chosen data, and sampling other long-context data from the rejected group of existing preference datasets as the new rejected data.

---

> ### Author Response · Authors · 2024-11-19
> **Response to Reviewer 6PbM (Part 1/2)**
>
> We greatly appreciate the reviewer's insightful comments and thorough feedback. Our detailed responses are provided below, and we hope they effectively address your concerns and contribute to a re-evaluation of our work.
>
> > Q1: not fully convinced by the motivation that learning from these paired preference data, in which the longer one is dispreferred enables the model to excel on long-context capabilities.
>
> A1: Thank you for your thoughtful question. We want to clarify that the core motivation behind our approach is to align a model's long-context performance with its proven short-context capabilities. **When dealing with extremely long contexts, such as those reaching 128K tokens, the relevant information pertinent to a query often constitutes only a small fraction of the entire context**. However, models without robust long-context abilities generally struggle to retrieve accurate responses when faced with such overwhelming context lengths.
>
> Our empirical data, as illustrated in the table below, highlights this limitation. In InfiniteBench tasks, where queries target information distributed in contexts up to 128K tokens, Mistral-7B-Instruct (with a context length of 32K) shows markedly improved performance when the context is truncated to these shorter lengths—even though this truncation entails significant information loss. This observation underscores that **LLMs typically falter in effectively leveraging long context as short context.**
>
> |     | Evaluation Length | En Sum | En QA | En MC |
> | --- | --- | --- | --- | --- |
> | Mistral-7B-Instruct | 128K | 22.13 | 4.93 | 14.41 |
> | Mistral-7B-Instruct | 32K by truncation | 23.30 | 14.22 | 44.98 |
>
> Hence, our motivation is **internally transfer the capabilities of LLMs within supported length to longer context lengths. If LLMs's performance on long context is as good as that on short context, we belive the LLMs are of remarkable long-context capabilities.** As known in the data construction process, the generated instructions are only related to a short chunk, so given the short chunk and instructions, the LLM can give desired responses as prefered ones. However, if we directly input the whole long document, the LLM may draw undesired behaviour and give dispreferred responses due to the overloaded context length. Hence, we appy the preference optimization between the two types of responses, seeking to make the LLM approach the capability on short context even when given the long context, i.e., **the implicit goal is the LLM can give the same quality of response despite given the short chunk or the whole long document. If so, the LLM is a well-performing long-context one.**
>
> > Q2: Haven’t provided evidence showing why LongPO is better than proper baselines, e.g., SFT with high-quality long-context data.
>
> A2: The reason why we haven't compared baseline of SFT with high-quality long-context data is that, **there are no open-source high-quality long-context SFT/DPO data** due to costly and even impractical long-context annotation. That said, the baseline GLM-4-9B in the paper involves training on private data based on human annotation on 128K length (with more steps like continual training), where the comparable performance in the table below between Mistral-7B-LongPO-128K and GLM-4-9B-128K further demonstrates the effectiveness of our LongPO.
>
> |     | Avg of Infinitebench | Avg of RULER |
> | --- | --- | --- |
> | GLM-4-9B-128K | 30.53 | 86.20 |
> | Mistral-7B-LongPO-128K | 39.27 | 86.06 |
>
> We hope this explanation clarifies our motivation and reinforces the efficacy of LongPO for enhancing long-context performance.
>
> > Q3: Is LongPO able to scale to much longer contexts?
>
> A3: We have extended the training length of LongPO to 512K on a set of 4.8K instances. The results, as presented in the table below, show an improvement in performance, suggesting that **LongPO has not yet reached its upper limit.**
>
> |                     | Train Length | En Sum | En QA | En MC | Avg of InfiniteBench | NIAH  | VT    | QA    | Avg of RULER |
> | ------------------- |:------------:| ------ |:-----:|:-----:|:--------------------:| ----- | ----- | ----- |:------------:|
> | Mistral-LongPO-128K | 128K         | 27.05  | 23.51 | 67.25 | 39.27                | 96.88 | 96.49 | 64.81 | 86.02        |
> | Mistral-LongPO-256K | 256K         | 28.16  | 24.43 | 66.35 | 39.65                | 96.80 | 97.0  | 64.87 | 86.22        |
> | Mistral-LongPO-512K | 512K         | 29.10  | 27.85 | 66.67 | 41.21                | 97.28 | 97.48 | 64.92 | 86.56        |
>
> However, **determining the exact point at which LongPO's performance plateaus remains challenging due to the absence of comprehensive benchmarks for contexts up to 1 million tokens and beyond.** The current longest evaluation in InfiniteBench, for instance, is only around 400K. As the field progresses, the development of more extensive evaluation criteria will be crucial to accurately assess LongPO's capabilities in long-context scenarios.

---

> > ### Author Response · Authors · 2024-11-19
> > **Response to Reviewer 6PbM (Part 2/2)**
> >
> > > Q4: Discuss the limitation of the proposed method.
> >
> > A4: **One limitation of LongPO, and other methods involving long-context full fine-tuning, is the infrastructure requirement for long-sequence training**. Long-sequence training relies on sequence parallelism such as DeepSpeed Ulysses (which we employed) and Ring Attention. However, the sequence parallel communication cost can be a bottleneck and significantly slow down training speed (e.g., train on 32K without sequence parallel may be 3x faster than train on 32K with sequence parallel). Additionally, DeepSpeed Ulysses has limitations in parallel degrees, restricting the maximum feasible training sequence length. Although Ring Attention can address this, it does so with slower speed and potential arithmetic errors, which may impact performance.
> >
> > **We look forward to future advancements in long-sequence training infrastructure and evaluation benchmarks to further unlock LongPO's potential.**
> >
> > > Q5: Some other ways of constructing preference data (e.g., sampling from existing datasets) ensuring one of chosen or rejected data has a long context, are able to achieve similar alignment performance as LongPO?
> >
> > Thank you for your insightful feedback. Preference optimization traditionally relies on **having two responses (chosen/rejected) to an identical query/instruction**, allowing the model to learn how to tailor its responses to more closely match the chosen example. Your suggestion raises an interesting possibility of leveraging simple strategies to generate preference data based on existing datasets, but there are some challenges.
> >
> > Firstly, when constructing preference data where one response is based on a long context and the other on a short context, **it is crucial to ensure that the short context is fully contained within the long context.** This ensures the comparability of the two responses by maintaining information equivalence. Without this, directly comparing the responses becomes problematic. **However, there are no existing datasets that meet such requirements**.
> >
> > Furthermore, creating long-context preference data from existing short-context datasets is **challenging due to the lack of available long-context preference/SFT data.** Previous studies[1] have indicated that simply assembling shorter contexts into a longer one does not necessarily result in improved long-context performance. This might be because the "long context" generated in this manner consists of semantically independent short contexts, which can be easily detected by LLMs, leading to suboptimal improvements in performance handling long contexts.
> >
> > Our LongPO utilizes LLMs to generate chosen/rejected responses given relative short/long documents respectively, hence circumventing the limitations above and eliminating the need for human annotation.
> >
> > [1] World Model on Million-Length Video And Language With Blockwise RingAttention

---

> ### Author Response · Authors · 2024-11-21
> **Request for Re-evaluation and Further Discussion**
>
> Dear Reviewers,
>
> Thank you for your valuable feedback. We submitted our responses over two days ago, addressing all your comments in detail. We kindly request that you review our rebuttal, re-evaluate our work, and engage in further discussion if you have any additional concerns.
>
> Best regards,
>
> Authors

---

> ### Author Response · Authors · 2024-11-24
> **Request for Response and Further Discussions**
>
> Dear Reviewer,
>
> As the author-reviewer discussion period approaches its end, we kindly request your feedback on our rebuttal. Your insights on whether we have effectively addressed your concerns would be greatly appreciated.
>
> We are truly grateful for the time and expertise you have dedicated to reviewing our work. Your insightful comments and suggestions have significantly contributed to enhancing the quality of our work.
>
> Thank you once again for your thoughtful feedback. We look forward to your further guidance and hope to address any remaining concerns before the discussion period closes.
>
> Warm regards,
>
> The Authors

---

> ### Comment · Reviewer_6PbM · 2024-11-25
> **Official Comment by Reviewer 6PbM**
>
> Dear Authors,
>
> Thank you for your response. My critiques have been largely addressed in your rebuttal. With that, I decide to increase my score to 6.

---

### Official Review · Reviewer_rbSY · 2024-10-26

**Soundness:** 2
**Presentation:** 2
**Contribution:** 2
**Rating:** 5
**Confidence:** 4

**Summary:**

This paper introduces a novel training method, LongPO, which can transfer the alignment capability on short-context to long-context scenarios.
By guiding the model to self-generate the training data and applying a DPO-based training objective, LongPO can greatly improve the model's performance on long-context tasks.
Experiments on the Mistral-7B model show promising results for the LongPO method.

**Strengths:**

The direction of this paper is correct; long-context alignment is essential for long-context models nowadays.

The method is simple yet efficient; LongPO transfers the model capability from short-context to long-context scenarios by utilizing the KL term.

**Weaknesses:**

**Method**:
I think the road of the overall approach is right, but the details should be improved. The method in this paper should be applied to the long-context models directly rather than doing both the context scaling and long-context alignment. The iterative self-evolving training is costly and complex, and it's better to directly use the positional extrapolation method (e.g., increase the value of RoPE base and conduct long-context post-training) to extend the context length and then utilize LongPO training to perform the long-context alignment.

**Experiment**: Lack of details for reproducing the experiment. For instance, what are the settings of SFT and DPO for Mistral in Table 1? How much data do you utilize? How long does the training take? There are many settings and details that should be reported since DPO-based methods are often difficult to implement.

**Writing**:
The derivation of RL formulas in the main body makes it difficult to understand the core idea the paper aims to present. I believe that if some of the derivations are included in the Appendix and only key formulas are retained in the main body, the readability will increase.

**Figure**:
There are too many marked arrows in Figure 2 without corresponding textual explanations. To be honest, this figure gives me a rough idea of what the author is doing, but the specific details are not clear.

**Questions:**

1. Why not utilize the open-source training data as the source data for generating the training data in your paper? I noticed that the authors utilize the Instruction Generation method to ensure the diversity of instructions. However, based on my experiments, the instructions generated from the model have a relatively low quality (especially from the Mistral-7B-Instruct-v0.2 model) compared to the open-source long-form instruction data, e.g., LongAlpaca, namespace-Pt data[1].

2. The experiments are mainly conducted on Mistral-7B-Instruct-v0.2, which already has a 32K context length. Why not conduct experiments on the short-context model (4K) like Llama-2 and the long-context model (200K) like Yi-200K?


[1] Here I provide the link for namespace-Pt data, which I think has a relatively high quality: https://huggingface.co/datasets/namespace-Pt/projects/resolve/main/long-llm.tar.gz

---

> ### Author Response · Authors · 2024-11-19
> **Response to Reviewer rbSY (Part 1/3)**
>
> We greatly appreciate the reviewer's insightful comments and thorough feedback. Our detailed responses are provided below, and we hope they effectively address your concerns and contribute to a re-evaluation of our work.
>
> > Strength: LongPO transfers the model capability from short-context to long-context scenarios by utilizing the KL term.
>
> Answer: Thanks for recognizing the simplicity and effectiveness of our method. However, sorry for any confusion, **the goal of transferring model's capability from short to long in LongPO is not achieved by utilizing the KL term**.
>
> We want to clarify more details about how LongPO works here. The two key components in LongPO are: (1) Short-to-Long Preference Learning introduced in Sec. 3.1; and (2) Short-to-Long Constraint (i.e., the KL term) introduced in Sec. 3.2.
>
> **The actual component to transfer the capabilities from short to long is the Short-to-Long Preference learning**. Given a long document and a query, the short-to-long preference is reflected between two responses: (1) a rejected response generated by the short-context LLM given the whole document and query; (2) a chosen response generated by the short-context LLM as well, but given a shorten document chunk that contains all related information for the query. So learning from these two responses through preference optimization can make the short-context LLM approach its short-context behaviour when processing long-context inputs, hence bringing forward the capabilities from short to long.
>
> **The KL term is utilized to maintain the short-context performance during long-context training**. RLHF and DPO both rely on a reference model to give a reference output distribution, to avoid deviating the original model output distribution during alignment based on the KL term in Eq (8). However, when trained on long context, as shown in Eq (8), the reference output distribution is given upon the $x_L$. **As the reference model is usually the original short-context model itself, it cannot provide good reference distribution due to limited long-context capabilities**. So we address this issue by introducing the short-to-long constraint KL term in Eq (10), that utilizes the short-context LLM to give the reference output distribution upon the short context $x_S$ that contains the ground truth answer for the query. In our ablation, we found this simple term can effectively maintain all short-context capabilities after long-context training.
>
> We hope the illustration above can help you have a clearer understanding of our method.
>
> > Q1: The method should be applied to the long-context models directly.
>
> A1: Thanks for your suggestions. LongPO addresses context scaling and long-context alignment in one stage **because our goal is to extend the context length of instruction-tuned LLMs without compromising their existing capabilities**. We also believe the ideal roadmap to develop a long-context LLM should be pretraining -> long-context continual pretraining -> long & short-context mixed SFT -> our LongPO ( to further improve the long-context performance while preserving the short-context capability).
>
> However, instruction-tuned LLMs, such as LLaMA 3-8B-Instruct, are often **released without detailed training recipes and access to alignment data**. That means, if we apply long-context continual training to extend the context length of LLaMA 3-8B-Base from 8K to 128K, it is non-trivial to achieve the same level of short-context alignment as LLaMA 3-8B-Instruct. If we apply long-context continual training on LLaMA 3-8B-Instruct, **the instruction-following capabilities would significantly drop and are hard to recover.**
>
> For example, Prolong-8B-512K-Base demonstrated a considerable decline in instruction-following capabilities after continual long-context training upon LLaMA 3-8B-Instruct (from 6.88 to 5.91 as measured by MT-Bench, using GPT-4-Turbo-1106 as the judge), with only minimal recovery after SFT (5.91 to 6.01).
>
> |     | MT-Bench |
> | --- | --- |
> | LLaMA 3-8B-Instruct | 6.88 |
> | ProLong-8B-512K-Base | 5.91 |
> | ProLong-8B-512K-Instruct | 6.01 |
>
> To the best of our knowledge, **our LongPO is the first one-stage method capable of extending the context length of instruct-tuned LLMs (e.g., Mistral-7B-Instruct), while preserving their short-context and instruction-following strengths (e.g., 6.35 for Mistral-7B-LongPO-128K vs 6.34 for Mistral-7B-Instruct on MT-Bench).** This capability addresses a critical gap in the current methodologies, ensuring that we can directly extend the context length of open-source instruct-tuned LLM. We hope the explanation above clarifies the significant contributions of our work in the long-context area.

---

> ### Author Response · Authors · 2024-11-19
> **Response to Reviewer rbSY (Part 2/3)**
>
> > Q2: Lack of details for reproducing the experiment?
>
> A2: Thank you for your feedback and we apologize for any confusion. Actually **we have included sufficient details for reproducing LongPO training in the initial submission**. In Sec. 4.1, we have listed that we use 45k of 128k-length documents for the iteration 1 training and 22K of 256K-length documents for the iteration 2 training. These documents are filtered from Book and Arxiv domains of Long-Data-Collection ([togethercomputer/Long-Data-Collections · Datasets at Hugging Face](https://huggingface.co/datasets/togethercomputer/Long-Data-Collections)) and GitHub domain of Redpajama ([togethercomputer/RedPajama-Data-1T · Datasets at Hugging Face](https://huggingface.co/datasets/togethercomputer/RedPajama-Data-1T)) according length, **which are all open-sourced**. All hyperparameters used for reproducing LongPO training are also listed (e.g., beta is set as 0.1, lambda is 0.01, learning rate is 5e-7, global batch size is 8) in Sec. 4.1. Based on Deepspeed-Ulysses sequence parallel, the throughput on 8xA800 is 4401 tokens/second. We train the model on 45K of 128K-tokens documents with QA pairs, taking around 372 hours. The memory usage is 582.27 GBs.
>
> In Section 5.1 of the initial version, we note that **SFT and DPO are trained using the same self-generated datasets as LongPO**. Since SFT cannot leverage preference data, it is applied to chosen responses alongside long context inputs. To enhance clarity, **we have added more details about training settings of SFT and DPO in Section B.3 of the Appendix.**
>
> We hope these details clarify your understanding of the LongPO training methodology. **All models and training data will be released after the ICLR review process.**
>
>
>
> > Q3: The derivation of RL formulas.
>
> A3: Sorry for any confusion. We are a bit confused about where the RL formulas are. Do you refer to the RLHF formula in Eq (1)? As we would reference the RLHF in Sec. 3.2, we need to list it in the preliminary section and have tried to make it concise. If you refer to the "KL" formulas in Eq (8) - (10) of Sec. 3.2, we may simplify them in the revised version of paper, i.e., put the Eq (9) into Appendix and omit some of the detailed derivations.
>
> > Q4: Too many marked arrows in Figure 2.
>
> A4: Thank you for bringing this to our attention, and we apologize for any confusion caused by Figure 2. The figure is intended to illustrate the short-to-long preference data curation process, and we appreciate your feedback on its clarity. Here's a detailed explanation of the steps depicted in Figure 2:
>
> 1. Instruction Generation (Black Dashed Arrows): The first step involves generating instructions. The model is prompted to create instructions based on a short context chunk extracted from a long document.
>
> 2. Short-Context Answer Generation (Pink Arrows): In the second step, the model generates a preferred response based on the instruction from step 1 and the short context chunk. This reflects the model's ability to respond accurately when given sufficient, focused context.
>
> 3. Long-Context Answer Generation (Green Arrows): The third step shows the model generating a dispreferred response using the entire long document and the same instruction from step 1. This demonstrates the challenges the model faces when processing overly extensive contexts.
>
> We now included the order of the arrow in the revised submission. We hope this explanation enhances your understanding of the short-to-long preference data curation process.

---

> ### Author Response · Authors · 2024-11-19
> **Response to Reviewer rbSY (Part 3/3)**
>
> > Q5: Why not utilize the open-source training data as the source data?
>
> A5:  First of all, we would like to clarify that the training data of LongPO sources from Long-Data-Collection (https://huggingface.co/datasets/togethercomputer/Long-Data-Collections) and Redpajama-Github (https://huggingface.co/datasets/togethercomputer/RedPajama-Data-1T) and both of them are publicly available (see Sec. 4.1 of the submission for more details).
>
> Regarding the generated instruction, we try to maintain its quality by prompting the experimented LLMs (e.g., Mistral-7B-Instruct-v0.2) with a short document chunk **instead of the whole long document** (please kindly refer to Fig. 2) and the generated instruction can be paired with either the short document chunk (as a synthetic short-context example) or the whole long document (as a synthetic long-context example). Although the generated instruction may not be as natural/good as the human-written instructions, such strategy allows for a fully automatic workflow for sythesizing long-context training data, which is critical when we want to extend to a very large context window (e.g., 10M).
>
> On the other hand, the comparison results between our LongPO and GPT-4 / LLaMA 3.1-8B suggest that the proposed short-to-long preference optimization framework  works well with synthetic instructions only. More specifically, **the effectiveness from self-generated instructions and responses further validates the self-evolution property of LongPO for extending LLMs' context lengths without external guidance.**
> That being said, we believe it will further benefit from the long-context datasets with high-quality instructions and are open to incorporating the suggested datasets (e.g., namespace-Pt-longllm) in future work.
>
> > Q6: The experiments are mainly conducted on Mistral-7B-Instruct-v0.2, which already has a 32K context length. Why not conduct experiments on the short-context model (4K) like Llama-2 and the long-context model (200K) like Yi-200K?
>
> A6: Thank you for your insightful suggestions. Our choice to conduct experiments primarily on the Mistral-7B-Instruct-v0.2, which has a 32K context length, is strategic to **facilitate comparisons with leading long-context language models like GPT-4-128K and LLaMA 3.1-128K on recent challenging benchmarks such as InfiniteBench (128K) and RULER (128K)**. Mistral-7B-Instruct-v0.2 allows us to efficiently extend the context length to 128K in a single iteration with LongPO, enabling a direct and practical comparison with these baselines to demonstrate the effectiveness of our proposed method.
>
> While LongPO is capable of extending a 4K model like LLaMA-2 to a 128K context length, it would require significantly more iterations and be less efficient for experimenting. On the other hand, Yi-200K has a context length beyond most existing LLMs and evaluation length of long-context benchmarks. Even if extending it to longer context length using LongPO, the resulting model is complicated to evaluate and compare with baselines, making the effectiveness hard to identify.

---

> ### Author Response · Authors · 2024-11-21
> **Request for Re-evaluation and Further Discussion**
>
> Dear Reviewers,
>
> Thank you for your valuable feedback. We submitted our responses over two days ago, addressing all your comments in detail. We kindly request that you review our rebuttal, re-evaluate our work, and engage in further discussion if you have any additional concerns.
>
> Best regards,
>
> Authors

---

> > ### Comment · Reviewer_rbSY · 2024-11-23
> > **Response to Authors**
> >
> > Dear Authors,
> >
> > Apologies for the delayed response. I have reviewed your rebuttal and have the following further questions and suggestions:
> >
> > ---
> > **1. Regarding Strength: LongPO transfers the model capability from short-context to long-context scenarios by utilizing the KL term.**
> >
> > I understood your explanation, thx for correcting me. However, I have a further question.
> > In your response, you mentioned, ``so learning from these two responses through preference optimization can make the short-context LLM approach its short-context behavior when processing long-context inputs``.
> > I am curious about which specific capabilities are transferred to long-context scenarios. Could you provide more concrete examples or a case study to illustrate this? For instance, could you demonstrate how LongPO addresses challenges posed by long-context inputs and how it leverages short-context behaviors to mitigate these challenges?
> >
> > ---
> >
> > **2. Regarding Question: Q1: The method should be applied to long-context models directly.**
> >
> > I understand that the objective of LongPO is to enable long-context models to acquire abilities akin to those of short-context models (although it is still unclear to me what those specific abilities are—see my question above). Based on this, I believe LongPO should not be conflated with context scaling methods. Instead, it seems to focus on long-context alignment, with context scaling being more of a byproduct. I may have misunderstood your approach, but given your writing structure, I would interpret LongPO as primarily a long-context alignment method rather than a context scaling technique.
> >
> > ---
> >
> > **3. Regarding Question: Q3: The derivation of RL formulas.**
> >
> > My concern pertains specifically to the RLHF formula in Eq. (1). I suggest including more experimental details in the main text, such as the training setup you referenced in your response to ``Q2 (Lack of details for reproducing the experiment)``. I believe these details would significantly benefit the long-context modeling field. Conversely, formulas like the "KL" equations in Eq. (8)–(10) of Sec. 3.2, which are relatively straightforward, could be summarized with their final conclusions in the main body while moving the derivations to the appendix.
> >
> > ---
> >
> > **4. Additional Concern: Ensuring the accuracy of model-generated data.**
> >
> > One aspect I previously overlooked is how you ensure the accuracy of model-generated data in your experiments. Even with short-context inputs, there is a possibility of model errors. How do you mitigate or account for this issue in your experiments?
> >
> > ---
> >
> > I hope these questions and suggestions help clarify my concerns.
> >
> > I look forward to your further explanations. Thank you!

---

> ### Author Response · Authors · 2024-11-23
> **Response to Reviewer rbSY (Second Round, Part 1/2)**
>
> We sincerely thank you for your thoughtful feedback on our previous responses. We have carefully addressed your additional concerns below and hope that this clarifies our position and positively influences your reconsideration of our submission.
>
> > Q1: which specific capabilities are transferred to long-context scenarios
>
> A1: The figure 1 in paper has displayed a real case (but we omit some content in the long document). When crucial informations are dispersed/obscured within a long document, the model may give a generally "right" (or wrong) answer but miss essential details (e.g, New York Times best-selling fiction list in the chosen response). This is often due to limited retrieval, tracing, and reasoning capabilities, which have been successfully transferred to longer context by LongPO as confirmed below:
>
> 1. **Retrieval and Muti-hop Tracing**: The NIAH and VT tasks in RULER involve retrieving key information (e.g., One of the special magic numbers for long-context is: 12345.) and tracing a multi-hop chain (e.g., X1 = 12345 ... Y1 = 54321...X2 = X1 ... X3 = X2 ...Find all variables that are assigned the value 12345), respectively. As can be seen in the table below, Mistral-7B-Instruct performs well on short context (16K) but gradually degenerates when the context length extends to 32K and 128K on both retrieval and tracing. However, our LongPO transfers the retrieval and muti-hop tracing capabilities to longer context, hence achieving concistent performance.
>
> |     | NIAH (16k) | NIAH (32k) | NIAH (128k) | VT (16K) | VT (32K) | VT (128K) |
> | --- | --- | --- | --- | --- | --- | --- |
> | Mistral-7B-Instruct | 94.3 | 85.5 | 7.4 | 92.2 | 85.0 | 0.0 |
> | - LongPO-128K | 98.09 | 97.65 | 91.18 | 98.08 | 96.36 | 93.28 |
>
>
> 2. **Reasoning over context**: this capability involves reasoning over some evidences within a context to get the answer, as seen in tasks of InfiniteBench. An example question from InfiniteBench is: “What color dress did A wear when A met B for the second time?” To further verify the transfering of this capability, we truncate the context (up to 128K) of En.QA and En. MC tasks in InfiniteBench **into 32K and 16K** and requery the models with a shorten context. The results below indicate the reasoning capability may degrade in Mistral-7B-Instruct when extending the context length, while our LongPO successfully transfers such capability to longer context (128K).
>
> |     | Evaluation Length | En QA | En MC |
> | --- | --- | --- | --- |
> | Mistral-7B-Instruct | 16K | 18.44 | 50.76 |
> | - LongPO-128K | 16K | 18.98 | 52.44 |
> | Mistral-7B-Instruct | 32K | 14.22 | 44.98 |
> | - LongPO-128K | 32K | 18.48 | 51.97 |
> | Mistral-7B-Instruct | 128K | 4.93 | 14.41 |
> | - LongPO-128K | 128K | 23.51 | 67.25 |
>
> These capabilities are crucial for LLM applications, such as long-duration and multi-turn dialogues or LLM agents. While LLMs typically retain these capabilities in short-context tasks, they may be lost or degraded in long-context scenarios. LongPO effectively addresses this issue, highlighting its effectiveness.
>
> ---
>
> > Q2: LongPO should not be conflated with context scaling methods.
>
> A2: Thanks for your insightful feedback. We appreciate and totally agree with your clear explanation that LongPO is "focusing on long-context alignment, with context scaling being more of a byproduct". The response in our rebuttal just aims to clarify that the byproduct may be also essential sometimes, as it is non-trivial to apply a separate context scaling phase for an instruct-tuned LLM.
>
> ---
>
> > Q3: Including more experimental details in the main text instead of the RLHF formula.
>
> A3: Thank you for your suggestion. Since it is not appropriate to make section-level changes during the rebuttal, we plan to move Section 2 to the appendix and incorporate additional information from Appendix B.3 into Section 4.1.
>
> ---

---

> ### Author Response · Authors · 2024-11-23
> **Response to Reviewer rbSY (Second Round, Part 2/2)**
>
> > Q4: Ensuring the accuracy of model-generated data.
>
> A4: Since the primary goal of our LongPO is to explore the self-evolving length generation property of the finetuned LLMs, we do not introduce advanced designs from external prior knowledge in filtering the model-generated data, but apply some basic methods to remove the outputs with undesired patterns, such as the censored outputs or the outputs with repetitive characters/words/phrases.
>
> To ease your concern about the quality of the model-generated data, we respectively prompt Mistral-7B-Instruct (the base model of our LongPO), GPT-4O-0806 and GPT-3.5-0614 with 20 sampled document chunks (in our training data) to generate instructions, and employ GPT-4-Turbo-1106 as a judge to assess the instruction quality, adopting the prompt template from [1]. As can be seen, even as a slightly outdated open-source LLM (compared with LLaMA 3.2 and Qwen 2.5), **Mistral-7B-Instruct can still generate instructions of reasonable quality (~80% of GPT-4O and ~87% of GPT-3.5-Tubo) without applying many filtering strategies.**
>
> We further query the three models to reply to instructions generated by Mistral-7B-Instruct based on corresponding document chunks. More surprisingly, the response quality score (also evaluated by GPT-4-Turbo-1106) of Mistral-7B-Instruct is even higher than that of GPT-3.5-Turbo-Instruct. This suggests that **the model is likely adequate for generating good responses to questions generated by itself in short-context scenarios**. However, **when generating responses based on entire long documents, the response quality score drops to 3.8**. These results validate the reasonable quality of self-generated data in LongPO training.
>
> |     | Instruction quality | Response Quality |
> | --- | --- | --- |
> | GPT-4O-0806 | 9.05 | 8.5 |
> | GPT-3.5-Turbo-Instruct | 8.4 | 7.35 |
> | Mistral-7B-Instruct-v0.2 | 7.3 | 7.5 |
>
> That being said, we believe LongPO will further benefit from high-quality data and would consider incorporating more advanced data filtering strategies in future work.
>
> [1] Judging LLM-as-a-Judge with MT-Bench and Chatbot Arena

---

> ### Author Response · Authors · 2024-11-25
> **[URGENT] need your immediate attention to our response**
>
> Dear Reviewer rbSY,
>
> We hope this message finds you well. The discussion period is ending soon, I am writing to emphasize the importance of your review for our submission. Your score is lower than the other two reviewers, which may suggest a possible misunderstanding or oversight.
>
> We have carefully addressed all concerns in our detailed rebuttal, and would greatly appreciate your timely review of it. A thorough reassessment is crucial to ensure a fair evaluation.
>
> Your expertise is highly valued, and we trust that a reconsidered review will reflect the true merit of our work.
>
> Thank you for your immediate attention to this matter.
>
> Best regards,
>
> Authors

---

> ### Author Response · Authors · 2024-11-27
> **[PDF Revision Period Ending Soon] Request for (If Any) Additional Questions**
>
> Dear Reviewer rbSY,
>
> Thank you for your valuable feedback on our paper. We are actively working on incorporating the revisions during this rebuttal phase into our PDF.  Please note that we **cannot update the PDF after November 27th.**
>
> We kindly request your feedback on whether our responses adequately address your concerns. **If you have any further questions or suggestions, please let us know so that we can address them and include the necessary revisions in the updated PDF.**
>
> Thank you once again for your insightful comments and for helping us improve our work.
>
> Best regards,

---

> > ### Comment · Reviewer_rbSY · 2024-11-28
> > **Rebuttal Feedback**
> >
> > Thanks for the authors' efforts in addressing my concerns.
> > I will maintain my score.

---

> > > ### Author Response · Authors · 2024-11-28
> > >
> > > Dear Reviewer,
> > >
> > > Thank you for your prompt response and for acknowledging our efforts to address your concerns. We value your feedback greatly, as it significantly contributes to enhancing our work.
> > >
> > > Could you kindly confirm if there are any remaining issues or concerns that you feel have not been adequately addressed? If so, we would appreciate it if you could provide further clarification on these points, as we are eager to make any additional improvements necessary.
> > >
> > > If your concerns have been fully addressed, we would be grateful if you might reconsider your current rating, especially as it is currently lower than that of the other reviewers.
> > >
> > > Thank you once again for your valuable input and support.
> > >
> > > Warm regards,
> > >
> > > Authors

---

### Official Review · Reviewer_FWUE · 2024-11-04

**Soundness:** 2
**Presentation:** 3
**Contribution:** 2
**Rating:** 6
**Confidence:** 3

**Summary:**

The paper proposes LongPO, a novel approach for extending LLMs' context length through self-evolution and preference optimization. The method leverages the model's existing short-context capabilities to generate preference pairs, using responses from short and long contexts as chosen/rejected samples. The approach incorporates a short-to-long KL constraint to maintain short-context performance while improving long-context handling. The authors demonstrate improvements over baseline Mistral-7B-Instruct, achieving competitive performance with larger models like GPT-4 on long-context tasks.

**Strengths:**

1. **Novel Framework:**
   - Introduces an innovative self-evolution approach for long-context adaptation
   - Presents a theoretically sound extension of DPO for long-context scenarios
   - Successfully maintains short-context performance during adaptation

2. **Technical Merit:**
   - Well-justified short-to-long KL constraint
   - Clear theoretical foundations and derivations
   - Effective integration of NLL loss for training stability

3. **Results:**
   - Achieves competitive performance with larger models
   - Demonstrates effective preservation of short-context capabilities
   - Shows promising results without external annotation

**Weaknesses:**

1. **Insufficient Comparisons:**
   - Lacks direct comparisons with recent long-context methods
    (e.g., "Effective Long-Context Scaling of Foundation Models", "LongLoRA")
   - No analysis against parameter-efficient approaches
   - Missing comparison with fine-tuning-free methods
    (e.g., "LLM Maybe LongLM: Self-Extend LLM Context Window Without Tuning")

2. **Limited Analysis:**
   - Insufficient analysis of computational costs and training efficiency
   - No investigation of potential combinations with existing methods
   - Limited exploration of different model architectures and scales

3. **Practical Considerations:**
   - Missing discussion of memory requirements
   - Limited analysis of inference-time computational overhead
   - No investigation of scaling behavior with different model sizes


**Suggestions for Improvement**
1. Include comprehensive comparisons with recent long-context adaptation methods
2. Add analysis of computational requirements and scaling behavior
3. Investigate potential combinations with existing approaches
4. Provide more detailed ablation studies
5. Include practical considerations for deployment

**Questions:**

1. How does LongPO compare with recent methods like LongLoRA in terms of computational efficiency and memory requirements?

2. Could the authors provide comparisons with fine-tuning-free approaches and analyze the trade-offs?

3. Have the authors explored combining LongPO with existing position embedding extension methods?

4. What is the computational overhead during inference compared to other long-context adaptation methods?

5. How does the method scale with different model sizes and architectures?

---

> ### Author Response · Authors · 2024-11-18
> **Response to Reviewer FWUE (Part 1/2)**
>
> We greatly appreciate the reviewer's insightful comments and thorough feedback. Our detailed responses are provided below, and we hope they effectively address your concerns and contribute to a re-evaluation of our work.
>
> > Q1: Lacks direct comparisons with recent long-context methods. (LLaMA-Long, LongLoRA)
>
> A1: We add the comparison between our LongPO and LongLoRA on InfiniteBench. Specifically, we instantiate LongLoRA with Llama-2-7b-longlora-32k and run the evaluations following the official settings of InfiniteBench (i.e., evaluation length of ~128K). Considering that the training length of Llama-2-7b-longlora-32k is 32K, we also truncate the inputs of each evaluation example to the length of 32K (i.e., evaluation length of 32K) and re-evaluate its performances for fair comparison. As can be seen, under both settings, LongLoRA cannot achieve the results comparable to our LongPO and it even fails on the inputs within its training length.
> |                         | Evaluation Length | En Sum | En QA | En MC |
> | ----------------------- | ----------------- | ------ | ----- | ----- |
> | Mistral-7B-Instruct     | 128K              | 22.13  | 4.93  | 14.41 |
> | Llama-2-7b-longlora-32k | 128K              | 4.34   | 0     | 0     |
> | Mistral-7B-LongPO-128K  | 128K              | 27.05  | 23.51 | 67.25 |
> | Mistral-7B-Instruct     | 32K               | 23.30  | 14.22 | 44.98 |
> | Llama-2-7b-longlora-32k | 32K               | 4.56   | 0     | 1.32  |
> | Mistral-7B-LongPO-128K  | 32K               | 26.27  | 18.48 | 51.97 |
>
> For LLaMA2-Long (i.e., the approach from "Effective Long-Context Scaling of Foundation Models"), despite the unavailability of the resultant models, we found that LLaMA 3.1 adopt exactly the same training strategies for long-context scaling (i.e., increasing the base frequency of RoPE to 500,000, long-context continual training on 800B tokens and SFT on synthetic data) and the comparison between LLaMA 3.1 and our LongPO has already been included in the Table 1 of our initial submission. According to the listed results, our LongPO-7B-128K **performs on par with** LLaMA3.1-8B-128K on Infinitebench (39.27 vs. 38.87), RULER (86.06 vs. 84.68) and Longbench-Chat (5.42 vs.6.22) at a **much lower training budget** (6B tokens vs. 800B tokens), demonstrating the effectiveness and the data efficiency of the proposed LongPO approach.
>
> > Q2: No analysis against parameter-efficient approaches.
>
> A2: Thank you for your suggestion. LongPO is designed to work with both full fine-tuning and parameter-efficient fine-tuning (PEFT) methods like LoRA and LongLoRA. In our experiments, we chose full fine-tuning due to its stable performance. To address your concern, we compared LongPO's performance and efficiency at 1000 training steps for full fine-tuning, LoRA, and LongLoRA on 8xA800 80GB, as shown in the table below. (Batch size per device is 1.)
>
>
> |                                     | En.Sum | En. QA | En.MC | Training Throughput (tokens/second) | GPU Memory (GB) |
> | ----------------------------------- |:------:|:------:| ----- |:---------------------------------------------:| --------------- |
> | Mistral-7B-Instruct                 | 22.13  | 4.93   | 14.41 | -                                             | -               |
> | Mistral-LongPO-FT (Step 1000)       | 26.40  | 19.79  | 62.88 | 4401                                          | 582.27          |
> | Mistral-LongPO-LoRA (Step 1000)       | 19.21  | 5.89   | 17.72 | 4625                                          | 349.68          |
> | Mistral-LongPO-LongLoRA (Step 1000) | 19.11  | 4.11   | 14.29 | 4650                                          | 348.29          |
>
>
> **In terms of efficiency, PEFT did not significantly outperform full fine-tuning under long-context (128K) conditions**. This is largely due to the sequence parallel technique (e.g., DeepSpeed Ulysses) required for long-context training, which splits sequences across multiple GPUs, creating communication bottlenecks. Consequently, the expected speedup from PEFT is minimal compared to full fine-tuning.
>
> Moreover, **PEFT still demands substantial memory for long-context training (128K)**. The memory needed for long-sequence activation, much larger than model parameters, necessitates distribution across GPUs, rendering single GPU training impractical. In our setup (per device batch size = 1, DeepSpeed Ulysses with Zero3), both methods require at least 8xA800 80GB GPUs, though PEFT uses less memory.
>
> Interestingly, LoRA and LongLoRA show a considerable performance gap compared to full fine-tuning in LongPO training. This suggests the limited trainable parameters in PEFT may restrict capability transfer in extreme long-context settings.
>
> **While this analysis provides initial insights, we consider the PEFT versus full fine-tuning discussion to be a broader research topic, distinct from our focus on LongPO.**

---

> ### Author Response · Authors · 2024-11-18
> **Response to Reviewer FWUE (Part 2/2)**
>
> > Q3: Missing comparison with fine-tuning-free methods.
>
> A3: We re-run Self-Extend with Mistral-7B-Instruct and verify its effectiveness on InfiniteBench. As can be seen, although the improvements of Self-Extend over vanilla Mistral are clear (+1.1 on En.Sum, +12.6 on En.QA, +20.5 on En.MC), it is still far from comparable to our LongPO (23.3 vs. 27.1 on En.Sum, 17.7 vs. 23.5 on En.QA, 34.9 vs. 67.3 on En.MC), which is merely trained on self-generated short-to-long preference data without external annotation.
> |               | En Sum | En QA | En MC |
> | ------------- | ------ | ----- | ----- |
> | Mistral-7B    | 22.13  | 4.93  | 14.41 |
> | - Self Extend | 23.27  | 17.67 | 34.93 |
> | - LongPO-128K | 27.05  | 23.51 | 67.25 |
>
> > Q4: Insufficient analysis of computational costs and training efficiency. && Missing discussion of memory requirements.
>
> A4: Thanks for your suggestion. We now list the training throughput of Mistral-7B-LongPO-128K in the table. Based on Deepspeed-Ulysses sequence parallel, the throughput on 8xA800 is 4401 tokens/second. We train the model on 45K of 128K-tokens documents with QA pairs, taking around 372 hours. The memory usage is 582.27 GBs. We hope this information provides a clearer picture of the resource demands and efficiency of our training process.
> |                        | Training Throughput (tokens/sec) | GPU Memory (GB) |
> | ---------------------- |:---------------------------------------------:|:---------------:|
> | Mistral-7B-LongPO-128K | 4401                                          | 582.27          |
>
>
> > Q5: No investigation of potential combinations with existing methods.
>
> A5: We would like to point out that our LongPO is essentially a post-training approach for long-context scaling and it is agnostic to existing methods, i.e., **the resulting models of LongPO are supposed to be compatible with arbitrary approaches (e.g., the training-free Self-Extend) in practice.** We conduct a group of preliminary experiments to apply Self-Extend to our Mistral-LongPO-128K and benchmark its performance on the Needle-in-a-Haystack (NIAH) test. Surprisingly, after applying Self-Extend, our Mistral-LongPO-128K can even perform reasonably on the sequences of 400K at no additional cost. We believe there is a great potential of combining LongPO with other approaches and we leave it for future work.
>
> |                                   | NIAH (128K) | NIAH (256K) | NIAH (400K) |
> | --------------------------------- | ----------- | ----------- | ----------- |
> | Mistral-7B                        | 10%         | 0           | 0           |
> | Mistral-LongPO-128K               | 100%        | 85%         | 0           |
> | Mistral-LongPO-128K + Self Extend | 100%        | 88%         | 44%         |
>
> > Q6: Limited exploration of different model architectures and scales. && No investigation of scaling behavior with different model sizes.
>
> A6: Thanks for your feedback. To address your concerns, we now include new results of LongPO on Qwen 2.5 across scales of 1.5B and 7B parameters in the table below.
> |                           | En Sum | En QA | En MC | Avg            |
> | ------------------------- | ------ | ----- | ----- | -------------- |
> | GPT-4                     | 14.73  | 22.44 | 67.25 | 34.81          |
> | Qwen2.5-1.5B              | 17.18  | 6.22  | 24.25 | 15.88          |
> | - LongPO-128K (Step 1000) | 23.04  | 14.67 | 47.60 | 28.44 (+12.56) |
> | Qwen2.5-7B                | 22.89  | 6.08  | 52.4  | 27.12          |
> | - LongPO-128K (Step 1000) | 31.52  | 17.04 | 66.34 | 38.3 (+11.18)  |
>
> As can be seen, LongPO significantly enhances the performance of the Qwen 2.5 series models. For instance, LongPO improves the Qwen2.5-7B to achieve an average score of 38.3 on the InfiniteBench, making it comparable to GPT-4 (34.81). Additionally, LongPO yields substantial improvements even on smaller models like Qwen2.5-1.5B by training on only 1000 steps (~1B tokens), with 12.56 absolute gains of average score.
>
> These results illustrate LongPO’s scalability and effectiveness across different model architectures and sizes, underscoring its general applicability and robustness.
>
> > Q7: Limited analysis of inference-time computational overhead.
>
> A7: We would like to clarify that our LongPO approach keeps the model architecture **completely unchanged** and there is **no additional overhead** during inference. That means the inference cost of LongPO-trained models is identical to that of the corresponding base LLMs. Taking Mistral-7B-LongPO-128K, which is trained from Mistral-7B-Instruct, as an example, its inference throughput for 128K context on a single A800 is around 14.6 tokens/second and the memory footprint is roughly 43 GB using bf16 precision.

---

> ### Author Response · Authors · 2024-11-21
> **Request for Re-evaluation and Further Discussion**
>
> Dear Reviewers,
>
> Thank you for your valuable feedback. We submitted our responses over two days ago, addressing all your comments in detail. We kindly request that you review our rebuttal, re-evaluate our work, and engage in further discussion if you have any additional concerns.
>
> Best regards,
>
> Authors

---

> ### Author Response · Authors · 2024-11-24
> **Request for Response and Further Discussions**
>
> Dear Reviewer,
>
> As the author-reviewer discussion period approaches its end, we kindly request your feedback on our rebuttal. Your insights on whether we have effectively addressed your concerns would be greatly appreciated.
>
> We are truly grateful for the time and expertise you have dedicated to reviewing our work. Your insightful comments and suggestions have significantly contributed to enhancing the quality of our work.
>
> Thank you once again for your thoughtful feedback. We look forward to your further guidance and hope to address any remaining concerns before the discussion period closes.
>
> Warm regards,
>
> The Authors

---

> > ### Comment · Reviewer_FWUE · 2024-11-25
> >
> > I appreciate the authors' thorough and thoughtful responses to the reviewer's comments. The inclusion of comparisons with recent long-context methods, as well as performance metrics from fine-tuning-free approaches, significantly addresses my initial concerns. The detailed analysis of computational costs and training efficiency provides valuable insights into the practical feasibility of the LongPO method.
> >
> > Moreover, the demonstration of LongPO's scalability across different model architectures and sizes, along with the lack of additional inference overhead, strengthens the case for its adoption in diverse contexts. Given these clarifications and the data provided, I am inclined to raise my score.

---

### Author Response · Authors · 2024-11-28
**General Response Regarding PDF Revisions**

Dear ACs, SACs, and Reviewers,

We are truly grateful for the insightful reviews and constructive feedback provided for our paper. It is heartening to see our work being acknowledged for its innovative self-evolution approach (Reviewer FWUE), essential investigation to the advancement of long-context LLMs (Reviewer rbSY), and remarkable effectiveness (Reviewer 6PbM). In response to the thoughtful suggestions from the reviewers, we have undertaken the following revisions:

- **Figure 1 Enhancement (Reviewer rbSY)**: We have revised Figure 1 by incorporating step order indicators along the arrows. This alteration clarifies the procedural workflow and improves the overall comprehensibility of the figure.

- **Addition Results on 512K length (Reviewer 6PbM)**: A new iteration (iteration 3) has been added to Table 1, extending the training length of LongPO to 512K with 4.8K training samples. The resulting improvements further demonstrate LongPO's capability to effectively extend the context length of LLMs beyond 512K, in alignment with Reviewer 6PbM's feedback.

- **Additional Results on more models (Reviewer FWUE)**: We present additional results for LongPO trained on Qwen 2.5-7B-Instruct for 3000 steps, following the same training settings used for Mistral-7B-LongPO-128K. The observed significant performance gains corroborate LongPO's robust effectiveness across various base models.

- **Efficiency and More Training Details regrading baselines (Reviewer FWUE)**: Appendix Section B.3 has been expanded to include detailed information on training efficiency and the SFT/DPO baselines, as well as the efficiency of our proposed method.

We appreciate the valuable feedback and the opportunity to refine our manuscript. We believe these revisions significantly enhance the clarity and impact of our contributions. Thank you once again for your invaluable input and guidance.

Best Regards,

Authors

---

### Public Comment · ~Guanzheng_Chen1 · 2025-02-20
**Camera Ready Revisions**

Dear ACs, SACs, and Reviewers,

We are sincerely grateful for your efforts during the review process. We have carefully incorporated the revisions from the rebuttal phase and are now submitting the camera-ready version of our paper. Below, we summarize the key updates:

1. We have expanded Section 4.1 and Appendix B.3 to provide a more comprehensive description of the data curation process and training setup.

2. We have added the experimental results for Qwen2.5-7B-Instruct, Qwen2.5-7B-LongPO-128K, Mistral-7B-LongPO-256K, and Mistral-7B-LongPO-512K to Table 1. Note that during the rebuttal phase, the evaluation length for Qwen2.5-7B and Qwen2.5-7B-LongPO-128K on RULER was calculated using the Mistral tokenizer. We have now turned this to use the Qwen2.5 tokenizer, aligning with the official setup where each model's evaluation length is calculated using its own tokenizer.

3. We list the numerical results of RULER and short-context tasks in Table 2 and 3, respectively.


Best Regards,

Authors

---

### Meta-Review · Area_Chair_b2yY · 2024-12-18

**Metareview:**

The paper proposes "LongPO" to improve LLM abilities for answering questions from long prompts, without the need for human annotation data. To do so, they create the following synthetic data generation pipeline:
1. Start with a base LLM which has problems handling long prompts but is competent at handling shorter prompts.
2. Use (maybe another) LLM to convert a long prompt $x_{L}$ into a shorter one $x_{S}$, provided that the shorter one still contains enough information to answer the question.
3. Obtain answers $y_{L}$ and $y_{S}$ from sending in $x_{L}$ and $x_{S}$ to the model, respectively.
4. Run DPO, conditioned on $x_{L}$, but with preference over $y_{S}$ (usually more accurate) than $y_{L}$. But add the KL term to make sure the model doesn't lose its abilities to solve short prompts either.

Experimental results mainly with Mistral-7B-instruct (with a few more models like Qwen 2.5-7B-instruct post-rebuttal) demonstrate that this:
* Allows drastic improvements on some long-context benchmarks (e.g. RULER), and can even be used at 512K token lengths, with no signs of plateauing yet. This outperforms leading models such as GPT4 as well.
* Ablation studies demonstrate that the KL constraint is needed, or else the model gets worse at short-context benchmarks.

The method is reasonable, and the authors provided substantial a substantial rebuttal to address most concerns. The results appear to be quite significant, and thus the paper should be accepted.

**Additional Comments On Reviewer Discussion:**

The reviewer scores were 5,6,6, making this paper borderline. Reviewer rbSY (who gave a 5) may have had a slight understanding (an issue raised by the authors) on the method of the work, and I generally agree.

As for the other reviewers (FWUE and 6PbM), they raised the following issues, along with author responses:
 * Method was not shown for LLM base models other than Mistral-7B, raising questions on its generalizability.
    * Authors then provided experiment for Qwen 2.5-7B-Instruct.
 * Comparisons against LongLoRA
   * LongLoRA is mainly an alternative gradient update method, which is orthogonal to the question of DPO. Authors provided results using LongLoRA + LongPO, and found it to be worse than regular gradient descent, due to LoRA's approximation errors / memory intensity.
 * Why DPO specifically is necessary over SFT
    * Authors claim that there are no long-context annotation data in the first place for SFT, so this is a moot point.
  * Extension to 512K tokens and beyond
    * Authors provided results demonstrating method still works for these lengths and does not plateau. It is reasonable that 1M+ token benchmarking was not done, due to lack of relevant benchmarks and supported base models.

---

### Decision · Program_Chairs · 2025-01-22

Accept (Poster)